# Ensemble Deep Learning for Real–Bogus Classification with Sky Survey Images

**DOI:** 10.3390/biomimetics10110781

**Published:** 2025-11-17

**Authors:** Pakpoom Prommool, Sirikan Chucherd, Natthakan Iam-On, Tossapon Boongoen

**Affiliations:** 1School of Applied Digital Technology, Mae Fah Luang University, Chiang Rai 57100, Thailand; pakpoom.pro@mfu.ac.th; 2Department of Computer Science, Aberystwyth University, Aberystwyth SY23 3FL, UK

**Keywords:** astronomical transients, Convolutional Neural Networks (CNNs), transfer learning, fine-tuning, ensemble learning, optical transient detection, biomimetics, bio-inspired computing

## Abstract

The discovery of the fifth gravitational wave, GW170817, and its electromagnetic counterpart, resulting from the merger of neutron stars by the LIGO and Virgo teams, marked a major milestone in astronomy. It was the first time that gravitational waves and light from the same cosmic event were observed simultaneously. The LIGO detectors in the United States recorded the signal for 100 s, longer than in previous detections. The merging of neutron stars emits both gravitational and electromagnetic waves across all frequencies—from radio to gamma rays. However, pinpointing the exact source remains difficult, requiring rapid sky scanning to locate it. To address this challenge, the Gravitational-Wave Optical Transient Observer (GOTO) project was established. It is specifically designed to detect optical light from transient events associated with gravitational waves, enabling faster follow-up observations and a deeper study of these short-lived astronomical phenomena, which appear and disappear quickly in the universe. In astrophysics, it has become more important to find astronomical transient events like supernovae, gamma-ray bursts, and stellar flares because they are linked to extreme cosmic processes. However, finding these short-lived events in huge sky survey datasets, like those from the GOTO project, is very hard for traditional analysis methods. This study suggests a deep learning methodology employing Convolutional Neural Networks (CNNs) to enhance transient classification. CNNs are based on how biological vision systems work and how they are structured. They mimic how animal brains hierarchically process visual information, making it possible to automatically find complex spatial patterns in astronomical images. Transfer learning and fine-tuning on pretrained ImageNet models are utilized to emulate adaptive learning observed in biological organisms, enabling swift adaptation to new tasks with minimal data. Data augmentation methods like rotation, flipping, and noise injection mimic changes in the environment to improve model generalization. Dropout and different batch sizes are used to stop overfitting, which is similar to how biological systems use redundancy and noise tolerance. Ensemble learning strategies, such as Soft Voting and Weighted Voting, draw inspiration from collective intelligence in biological systems, integrating multiple CNN models to enhance decision-making robustness. Our findings indicate that this bio-inspired framework substantially improves the precision and dependability of transient detection, providing a scalable solution for real-time applications in extensive sky surveys such as GOTO.

## 1. Introduction

The detection of the fifth gravitational wave, known as GW170871 [1] and its electromagnetic signals—which were brought on by the merging of neutron stars—were announced by the team of scientists from the LIGO and Virgo laboratories. Astronomy has reached a major turning point with this. These gravitational waves were detected for over 100 s by the two LIGO gravitational wave detectors in Hanford and Livingston, USA, which is longer than the signals seen in the four preceding occurrences. Usually, huge objects like neutron stars or black holes combine to produce gravitational waves. The merger creates a longer-lasting gravitational wave signal if it involves two neutron stars, which are tiny and incredibly dense. Electromagnetic waves of every spectrum, from radio waves to gamma rays, are released during the merging. Unfortunately, the sophisticated LIGO and Virgo detectors are not very good at identifying the precise origin of gravitational waves. The gravitational wave signal must be found by rapidly scanning and searching a wide region of the sky. There is a higher chance of analyzing the transitory event if the source of the gravitational wave is discovered faster. The Gravitational-Wave Optical Transient Observer (GOTO) [2,3] project was conceived and designed in part to address this difficulty. There is more opportunities to use follow-up observatories and satellites to examine the short-lived source and its surroundings if possible electromagnetic (EM) equivalents can be found more quickly after gravitational waves are detected. While this problem was less difficult for the GW170871 event because of its very accurate localization, the projected search region has been significantly greater for many subsequent occurrences. The Gravitational-Wave Optical Transient Observer (GOTO) telescope project was specifically designed to find and detect optical phenomena related to gravitational wave observations. The observation of transient events—brief occurrences that arise and vanish rapidly—is now possible because of major advancements in optical telescope technology in recent years. The term “transient” describes astronomical occurrences that last for a short while before disappearing.

HOTPANTS [4] is a long-standing and effective image subtraction tool used in astronomy to detect transient objects and brightness variations. However, it has key limitations affecting accuracy and flexibility. Over 50% of small stars often remain after subtraction because a single kernel cannot handle varying star sizes. Parameter tuning is also manual and non-generalized, requiring repeated adjustments of Gaussian functions and σ values for different datasets. Additionally, HOTPANTS performs poorly on high-noise images or when PSF differences are large, leading to incomplete subtraction and residual artifacts, and it is computationally slow for large surveys like GOTO and ZTF. To overcome these issues, researchers now employ deep learning approaches—particularly Convolutional Neural Networks (CNNs)—which can automatically learn object and background features without manual tuning, significantly improving subtraction accuracy and transient detection in real observations.

In modern astrophysics, detecting astronomical transients [5,6]—such as supernovae, gamma-ray bursts, and stellar flares—is crucial for understanding high-energy cosmic events like neutron star mergers and black hole collisions, which reveal fundamental physical laws beyond laboratory replication. However, identifying these short-lived phenomena is challenging, especially in large-scale surveys like GOTO, which captures over 400 sky images nightly, each with more than 20,000 celestial objects [7]. Manual inspection is impractical, necessitating AI systems capable of biologically inspired learning and decision-making.

The concept of biomimetics underpins this study, combining mechanisms inspired by natural intelligence. CNNs mimic hierarchical visual processing in the brain [8], while transfer learning emulates adaptive knowledge transfer by fine-tuning pretrained models (e.g., VGGNet, ResNet, Inception, and Xception) for small or imbalanced astronomical datasets [9]. Data augmentation (rotation, flipping, and noise) models environmental adaptation [10], and dropout regularization imitates biological redundancy to prevent overfitting [11,12]. Inspired by swarm intelligence, ensemble learning (Soft and Weighted Voting) integrates multiple CNNs for more stable and noise-tolerant predictions [13,14,15].

This bio-inspired framework has achieved high accuracy and robustness in classifying transients, making it suitable for real-time alert systems like GOTO. Future work includes using Generative Adversarial Networks (GANs) to create synthetic data for underrepresented classes, mimicking the brain’s imagination process. As shown in [16,17], GAN-based augmentation enhances the classification of rare variable stars. Overall, this integrated biomimetic approach—combining visual learning, adaptive generalization, ensemble decision-making, and synthetic data generation—offers a scalable and precise solution for next-generation astronomical surveys.

The main contributions of this study can be summarized as follows:We propose an ensemble deep learning framework that integrates multiple pretrained CNN architectures to enhance accuracy and robustness in astronomical transient detection.We evaluate both transfer learning and fine-tuning strategies under diverse data augmentation settings (Original, Rotation, Noise, HFlip, and VFlip) and batch sizes ranging from 32 to 256.We demonstrate that the proposed ensemble approach significantly reduces false positives and improves detection reliability compared with individual CNN models.We analyze the architectural differences among the best-performing models (e.g., depthwise separable convolutions in MobileNet and Inception-style modules in Xception) to explain why specific networks perform better under certain augmentation conditions.We provide insights that contribute to the development of scalable and generalizable deep learning solutions for real–bogus classification in wide-field sky surveys such as GOTO.

The remainder of this study is organized as follows: Section 2 presents the materials and methods, including the system overview, dataset, data acquisition process, modeling procedures, and evaluation methods. Section 3 reports the experimental results, consisting of preliminary experiments using various models under different data augmentation settings, as well as the main experiment based on the proposed ensemble deep learning approach. Section 4 provides an in-depth discussion of all experimental results presented in the previous sections. Section 5 presents the conclusions of this study, summarizing the key findings and implications. Section 6, Future Work, outlines potential directions for extending and improving this research. 

## 2. Materials and Methods

### 2.1. System Overflow

In Figure 1, nine deep learning models—DenseNet121, InceptionV3, MobileNet, MobileNetV2, ResNet101, ResNet50, VGG16, and VGG19—were selected to evaluate their performance. The original data in FITS format were normalized to a range between 0 and 1 and expanded using four data augmentation techniques—noise, rotation, vertical flip (VFlip), and horizontal flip (HFlip)—to increase data diversity and improve model generalization.

After data augmentation, the expanded dataset and original images were used to train all nine models using the transfer learning approach with pretrained ImageNet weights. The data were divided into training, validation, and testing sets. Each model was then fine-tuned to enhance the performance of deeper layers unaffected by the transfer process. After training and fine-tuning, each model’s performance under each augmentation type was evaluated using the validation set, and the best-performing model from each group was selected for the final stage—ensemble deep learning.

### 2.2. Dataset

In Figure 2, the dataset shows a transient discovery image, divided into two parts, real and bogus, divided by astronomy experts; both real and bogus images are 21 × 21 pixels. There are 523 real images and 3598 bogus images [18].

#### 2.2.1. Data Acquisition

In this preliminary study, we focused on analyzing simulated images generated for the GOTO telescope instead of using real observational data. The advantage of simulated data lies in knowing true transient sources beforehand, making it ideal for testing supervised machine learning methods, though it may not fully capture the complexity of real observations. The simulated images were created using SkyMaker vesrion 3.3.3 [19], which reproduces typical observational effects such as background noise and the Point Spread Function (PSF). The source list was compiled from two catalogs—SDSS for faint stars and galaxies and UCAC for bright stars (magnitude < 17)—combining both to extend the dynamic range. Based on the telescope’s field of view, each source’s RA/Dec was converted into pixel coordinates using a scale of 1.24 arcseconds per pixel, with G-band magnitudes from SDSS and V-band magnitudes from UCAC. Galaxies were modeled as disks only, omitting bulges, since the study focused on transient sources. SkyMaker required a configuration file, defining instrument properties such as the photometric zeropoint (23.5), PSF size, CCD dimensions (8176 × 6132 pixels), and pixel saturation level (65,535). The PSF FWHM was randomly varied between 0.8 and 3 arcseconds to increase realism. Two images were generated for each sky field, with new sources of magnitude 14–19 added randomly in the second image to simulate transient events. All simulated images were processed using a modified LSST software stack version 21.0.0 [20], and the image differencing outputs were then used as inputs for the machine learning algorithms.

#### 2.2.2. Data Preparation

The dataset used in this study comprises 4101 instances, consisting of 523 labeled as “real” and 3578 labeled as “bogus.” The data were randomly split into three subsets. For each subset, the training set contained 3280 samples (2862 “bogus” and 418 “real”), while the testing set contained 821 samples (716 “bogus” and 105 “real”). Notably, the “real” data also included simulated instances. Figure 2 presents several representative examples in image format. The original data were in FITS format, and they were subsequently normalized to a range of 0 to 1 using the equations shown below. Here, *arr* refers to the array of one image, arr1 is obtained by subtracting the minimum value of *arr* from each element in *arr*, and the final result (*arr*2) is obtained by dividing *arr*1 by the maximum value of *arr*1.(1)$$arr1=arr-{arr}_{min}$$(2)$$arr2=\frac{arr1}{{arr1}_{max}}$$

In addition, the dataset exhibits a clear class imbalance, where the number of “real” images is significantly smaller than that of the “bogus” images, at an approximate ratio of 1:7, as shown in Table 1. To address this issue, oversampling techniques were applied to increase the number of training samples in both classes using data augmentation. Each augmentation method (e.g., noise injection, image rotation, horizontal flipping, vertical flipping) was applied independently to avoid the confounding effects of combined transformations. Finally, all images were resized to 224 × 224 pixels to ensure compatibility with the input requirements of ImageNet-based architectures during the transfer learning process.

### 2.3. Data Augmentation

Data augmentation plays a crucial role in deep learning [21,22], as it increases data diversity without needing additional real-world observations—an advantage in astronomy, where data collection is limited by time and resources [23]. By transforming images in different ways, the dataset grows, reducing overfitting and improving generalization. In this study, several augmentation methods were used to match the characteristics of astronomical images. Noise injection adds random “salt-and-pepper” noise to simulate sensor errors, weather effects, or background noise, helping the model detect real sources under poor image quality. Rotation (e.g., 90° or 180°) mimics changes in telescope orientation, while vertical (VFlip) and horizontal flipping (HFlip) simulate variations caused by different imaging angles. These transformations make the model invariant to position, orientation, and noise, thereby enhancing accuracy, robustness, and adaptability in astronomical image classification. In Figure 3 show an example of data augmentation. 

### 2.4. Modeling

Convolutional Neural Networks (CNNs) [24,25] are powerful deep learning tools for image classification because they learn hierarchical spatial features, making them highly effective at identifying celestial objects through structural and light distribution analysis. Over the past decade, various CNN architectures have been developed, each differing in depth, parameters, and computational efficiency. In this study, we selected nine architectures to evaluate their performance in classifying astronomical transient images: VGG16, VGG19, ResNet50, ResNet101, InceptionV3, Xception, DenseNet121, MobileNet, and MobileNetV2. Since deep learning typically requires large datasets, the limited sample size posed a challenge. To address this, transfer learning was used to leverage pretrained ImageNet weights for efficient feature extraction and better generalization. This experiment thus serves as a preliminary adaptation of ImageNet-based transfer learning for sky survey images in transient detection, conducted on a system with an Intel Core i9-10900 K CPU, RTX 2080 Ti GPU, and 64 GB DDR4 RAM.

These architectures consist of the following:DenseNet121 [26]: Utilizes a dense connectivity mechanism, where each layer receives input from all preceding layers. This promotes feature reuse and alleviates the vanishing gradient problem.InceptionV3 [27]: Employs factorized convolutions and efficient dimensionality reduction, enabling deeper networks with lower computational cost.MobileNet [28]: Designed for mobile and embedded systems, this architecture uses depthwise separable convolutions to significantly reduce computational complexity.MobileNetV2 [29]: An extension of MobileNet, this version introduces inverted residual blocks, enhancing learning capacity while maintaining model compactness.ResNet50 and ResNet101 [30]: Implement shortcut connections or identity mappings to combat the vanishing gradient issue and enable effective training of very deep networks.VGG16 and VGG19 [31]: Feature a simple and sequential architecture composed of stacked convolutional layers with fixed kernel sizes, known for their consistency and reliability.Xception [32]: Evolved from the Inception architecture by replacing all modules with depthwise separable convolutions, offering improved efficiency in extracting fine-grained features.

All nine CNN architectures were trained and fine-tuned using the same hyperparameters (Table 2). Four batch sizes—32, 64, 128, and 256—were tested to evaluate their impact on convergence and generalization. Smaller batches (e.g., 32) generally improve generalization, while larger ones may overfit [33,34]. Training was limited to 100 epochs with Early Stopping (patience = 3) to prevent overtraining [35]. The Adam optimizer was used for its adaptive learning rate and fast convergence in noisy conditions [36]. The binary cross-entropy loss function was applied since the task involves distinguishing between two classes: real and bogus. The initial learning rate for the transfer learning phase was set to 0.001, while a lower rate of 0.00001 was used during fine-tuning to ensure stable updates in deeper layers and preserve pretrained features [37,38]. During fine-tuning, only the top 30% of convolutional layers were unfrozen and retrained to adapt domain-specific features from astronomical transient images, following best practices for moderate domain shifts and small datasets [39,40]. To ensure fairness, these hyperparameter settings were applied consistently across all models.

### 2.5. Transfer Learning

Transfer learning enhances model performance on a new but related task by reusing knowledge learned from a previous one. Models pretrained on large datasets like ImageNet [41,42,43] (over 14 million images in 1000 classes) capture general visual features such as edges, contours, and textures, which can be effectively reused for smaller or domain-specific datasets [44]. In this study, CNN architectures pretrained on ImageNet were adapted to classify astronomical transient images (Figure 4). The original classification head was replaced with a new fully connected layer for binary classification of real and bogus objects. Two training strategies were applied: transfer learning, where only the new classifier was trained while convolutional layers remained frozen, and fine-tuning, where the top 30% of convolutional layers were unfrozen and retrained with the classifier to capture domain-specific patterns [45,46,47]. This approach efficiently reuses existing knowledge, improves generalization, and enables effective domain adaptation for transient detection.

### 2.6. Ensemble Deep Learning

In the final stage of this study, ensemble deep learning [48,49,50] was employed to enhance classification accuracy and model stability when dealing with diverse astronomical images. As shown in Figure 5, five CNN models were selected and trained using different data augmentation strategies [51]—original, rotation, horizontal flip (HFlip), vertical flip (VFlip), and noise injection—with varying batch sizes (32, 64, 128, 256). Architectures such as MobileNet and Xception were chosen based on their strong validation performance in both transfer learning (TL) and fine-tuning (FT) phases. The ensemble combined predictions from all models through a voting mechanism, allowing each specialized model to contribute its strengths—for instance, rotation-trained models excel in detecting orientation changes, while noise-trained models handle low-signal-to-noise data. Prior research shows that ensemble voting improves robustness and generalization compared with single-model systems [52]. This approach is particularly effective in astronomy, where image variability in brightness, morphology, and observation conditions makes ensemble learning essential for achieving accurate and reliable transient classification.

### 2.7. Evaluation Methods

The experimental results from each route were collected and compared to evaluate performance and efficiency. In deep learning-based scientific classification, selecting proper evaluation metrics is crucial. Key indicators—accuracy, precision, recall, and F1-score—were used as outlined in Table 3. Here, TP, TN, FP, and FN represent correctly or incorrectly predicted positive and negative cases. Since accuracy alone can be misleading for imbalanced datasets [53,54] precision, recall, and, especially, the F1-score—the harmonic mean of precision and recall—were emphasized, as this provides a balanced assessment when both false positives and false negatives carry significant impact [55].

## 3. Results and Discussion

At this point, we compare how well each model and each data augmentation technique works with different types of datasets to see how well they function. To find the best models, we look at key performance metrics like accuracy, precision, recall, and F1-score. The results of this comparison will help us choose the best models to use in the next deep ensemble learning experiments. These experiments will combine the best features of different architectures to achieve better accuracy, robustness, and reliability in classifying astronomical transients under different noise and distortion conditions.

### 3.1. Comparison of Classification Results of Different Deep Learning Techniques with Original (Nonaugmented) Dataset

A comparative experiment using the original (non-augmented) astronomical dataset evaluated transfer learning (TL) and fine-tuning (FT) across four batch sizes (32, 64, 128, and 256) and nine CNN architectures: DenseNet121, InceptionV3, MobileNet, MobileNetV2, ResNet50, ResNet101, VGG16, VGG19, and Xception. The results showed that most models performed well, with MobileNet and VGG16 being the most stable and accurate. The fine-tuned MobileNet (batch 64) achieved the highest performance—accuracy = 0.98938 and F1-score (real) = 0.95758—maintaining strong results even with larger batches. DenseNet121 and Xception also achieved high accuracy (≈0.98) and solid F1-scores, demonstrating effective deep feature extraction. VGG16 and VGG19 performed consistently well despite their depth, while ResNet50/101 showed unstable precision and F1-scores, likely due to class imbalance. Overall, fine-tuned MobileNet (batch 64) was the best model, combining high accuracy, efficiency, and low computational cost. These findings highlight the importance of selecting suitable architectures, tuning batch sizes, and optimizing training strategies to achieve robust classification even with non-augmented astronomical data. In Figure 6 illustrates the accuracy and loss graph of different deep learning techniques with original (nonaugmented) dataset and Table 4. Represents the comparison of classification results of different deep learning techniques with original dataset.

### 3.2. Comparison of Classification Results of Different Deep Learning Techniques with Rotation Dataset

In this experiment, only rotation-augmented astronomical images were used to improve the models’ ability to recognize rotated objects. Two learning methods—transfer learning (TL) and fine-tuning (FT)—were tested across nine CNN architectures (DenseNet121, InceptionV3, MobileNet, MobileNetV2, ResNet50, ResNet101, VGG16, VGG19, Xception) and four batch sizes (32, 64, 128, and 256). The results showed that Xception achieved the best overall performance, with TL (batch 128) yielding accuracy = 0.97750 and F1-score = 0.97761, though FT (batch 256) also maintained high stability. VGG16 (FT, batch 128) and VGG19 (FT, batch 256) followed closely with strong and consistent results. The MobileNet and ResNet models, however, suffered from overfitting or class imbalance—especially ResNet50 (FT, batch 256), which failed completely. Smaller batch sizes improved ResNet101 stability, while MobileNetV2 (TL, batch 256) achieved competitive performance with low computational cost.

In summary, rotation-based augmentation proved highly effective when paired with suitable architectures and batch sizes. The best-performing models—Xception (TL, batch 128), VGG16 (FT, batch 128), and VGG19 (FT, batch 256)—demonstrated superior accuracy and F1-scores across both “real” and “bogus” classes, emphasizing the importance of proper model selection, hyperparameter tuning, and regularization for optimal results. In Figure 7 illustrates the accuracy and loss graph of different deep learning techniques with rotation dataset and Table 5. Represents the comparison of classification results of different deep learning techniques with rotation dataset. 

### 3.3. Comparison of Classification Results of Different Deep Learning Techniques with Noise Dataset

In this experiment, all models were trained on astronomical images with added noise to simulate real observational conditions, such as sensor artifacts, blurring, and low light. Nine CNN architectures—DenseNet121, InceptionV3, MobileNet, MobileNetV2, ResNet50, ResNet101, VGG16, VGG19, and Xception—were evaluated using transfer learning (TL) and fine-tuning (FT) across four batch sizes (32, 64, 128, and 256). Overall, most models struggled to learn from noisy data, with MobileNet, MobileNetV2, VGG16, VGG19, ResNet50, and ResNet101 showing an accuracy of 0.50000 and F1-scores of 0.00000, indicating complete failure in distinguishing real and bogus classes. Only a few models performed above random chance, notably, Xception and InceptionV3. The fine-tuned Xception (batch 256) achieved the best results with accuracy = 0.72625 and F1-score (bogus) = 0.76776, while ResNet50 (FT, batch 256) followed with accuracy = 0.85000 and F1-score (real) = 0.83827. InceptionV3 (TL, batch 32) showed moderate learning ability (accuracy = 0.63187; F1-score (real) = 0.43092) but tended to overfit under fine-tuning. Other models showed unstable or misleading results, such as MobileNetV2 (TL, batch 256), which had high precision but extremely low recall. In summary, adding noise significantly degraded model performance across most architectures. Xception (FT, batch 256) was the most noise-tolerant, followed by ResNet50 (FT, batch 256) and InceptionV3 (TL, batch 32). These findings highlight the need for denoising preprocessing, noise-aware training, and hybrid augmentation pipelines to improve model robustness and generalization under noisy astronomical conditions. In Figure 8 illustrates the accuracy and loss graph of different deep learning techniques with noise dataset and Table 6. Represents the comparison of classification results of different deep learning techniques with noise dataset. 

### 3.4. Comparison of Classification Results of Different Deep Learning Techniques with HFlip Dataset

In this experiment, horizontal flip (HFlip) was applied to expand all datasets by mirroring astronomical images, allowing CNNs to recognize objects viewed from reversed telescope angles. Nine architectures—DenseNet121, InceptionV3, MobileNet, MobileNetV2, ResNet50, ResNet101, VGG16, VGG19, and Xception—were tested using transfer learning (TL) and fine-tuning (FT) across four batch sizes (32, 64, 128, and 256). The results showed that most models, particularly Xception, MobileNet, InceptionV3, VGG16, and VGG19, achieved nearly perfect performance. Xception reached accuracy = 0.99750–0.99813 and F1-score = 0.99688–0.99875, while MobileNet (TL, batch 128) achieved accuracy = 0.99875 and F1-score = 0.99875, matching VGG19 (FT). InceptionV3 consistently maintained accuracy > 0.996 and F1 > 0.99, and DenseNet121 (TL) also performed strongly (F1 > 0.993). However, MobileNetV2 (FT) showed moderate results (accuracy: 0.91–0.92; F1: 0.91–0.93), while ResNet50 and ResNet101 failed under some FT setups (accuracy = 0.5; F1 = 0.0). Overall, HFlip significantly improved model learning for symmetric and mirrored features, with deep, well-structured models like Xception, MobileNet, and InceptionV3 performing best. Both FT and TL delivered high accuracy, proving HFlip to be an effective augmentation method for astronomical image classification. In Figure 9 illustrates the accuracy and loss graph of different deep learning techniques with original (nonaugmented) dataset and Table 7. Represents the comparison of classification results of different deep learning techniques with hflip dataset.

### 3.5. Comparison of Classification Results of Different Deep Learning Techniques with VFlip Dataset

In this experiment, the Vertical Flip (VFlip) technique was applied to expand the dataset by flipping astronomical images vertically, enabling models to recognize objects captured in inverted orientations caused by varying telescope angles. Nine CNN architectures—DenseNet121, InceptionV3, MobileNet, MobileNetV2, ResNet50, ResNet101, VGG16, VGG19, and Xception—were tested with transfer learning (TL) and fine-tuning (FT) using batch sizes of 32, 64, 128, and 256. The results showed that well-structured, deeper models like MobileNet, Xception, InceptionV3, VGG16, and VGG19 performed excellently, achieving near-perfect accuracy, precision, recall, and F1-scores for both classes. MobileNet (TL) and VGG19 (FT) reached 0.99875, while Xception (TL) achieved 0.99813, maintaining high stability even with large batch sizes. DenseNet121 and InceptionV3 also demonstrated strong, consistent results, whereas ResNet50 and ResNet101 struggled under some fine-tuning conditions but improved with TL. Overall, VFlip augmentation proved highly effective for models such as Xception, MobileNet, InceptionV3, VGG19, and VGG16, particularly with medium batch sizes (64–128), enhancing model robustness against variations in image orientation during real astronomical observations. In Figure 10 illustrates the accuracy and loss graph of different deep learning techniques with vflip dataset and Table 8. Represents the comparison of classification results of different deep learning techniques with vflip dataset.

### 3.6. The Classification Performance of Ensemble Deep Learning

In this study, various CNN architectures were trained and tested under different conditions using data augmentation, transfer learning (TL), fine-tuning (FT), and varying batch sizes to assess how training methods affect astronomical image classification in complex scenarios. The results showed that Xception, MobileNet, InceptionV3, VGG16, and VGG19 consistently achieved high accuracy, precision, recall, and F1-scores across augmented datasets (original, vertical flip, horizontal flip, and rotation), demonstrating strong adaptability to spatial and symmetrical variations. Table 9 represents selected models for ensemble learning based on best performance by augmentation type.

However, the main challenge was noise sensitivity. When noise augmentation was applied to simulate real observational conditions, most models showed significant performance drops, with some reaching F1 = 0.00000 for the “real” class—indicating complete failure in identifying true astronomical objects. These findings highlight model bias and limited generalization under noisy conditions. The best-performing models for each augmentation type were then selected based on both accuracy and F1-score, considering the optimal batch size for each case.

In every experiment, we used an entirely new test set that the models had never seen before. The test set consisted of 4000 real images and 4000 bogus images. However, for the experiment using the original data, we used a smaller set comprising 105 real and 715 bogus images, which were separated from the original dataset. The models had also never encountered these images during training. Overall, each experiment was evaluated using a distinct test dataset corresponding to its respective data augmentation type.

#### 3.6.1. Experimental Results of Model Combination Using Soft Voting Ensemble Technique

From Table 10, Table 11, Table 12, Table 13 and Table 14 show the results show that the Soft Voting ensemble method performs effectively across multiple datasets—original, rotation, noise, HFlip, and VFlip. The model achieved its best performance on the original dataset (accuracy = 0.9937, F1 (real) = 0.9720), correctly classifying nearly all samples with only six errors out of 820 images. Performance declined with rotation (accuracy = 0.7386, F1 = 0.6499) and dropped sharply with noise (accuracy = 0.5006, F1 = 0.0024), indicating difficulty recognizing rotated or noisy patterns. By contrast, the HFlip and VFlip datasets maintained excellent results (HFlip accuracy = 0.9867, F1 = 0.9860; VFlip accuracy = 0.9890, F1 = 0.9888), showing strong learning for symmetrical spatial features. Overall, the Soft Voting ensemble excels on datasets preserving spatial consistency (original, HFlip, and VFlip) but struggles with rotated and noisy data. The next phase explores noise-handling techniques and Weighted Voting ensembles to improve robustness and adaptability in realistic astronomical imaging conditions.

#### 3.6.2. Experimental Results of Model Combination Using Weighted Voting Ensemble Technique

We began the experiment by giving the model the weights it needed to start. After that, we looked at the overall accuracy to see how well it worked as a group. If a certain type of data augmentation made the accuracy go down, its weight was raised to make it more important for the ensemble to vote on. We checked the performance again to see how the change in weight affected the overall results of the classification. This step-by-step method helped us to slowly make the ensemble more balanced, ensuring that models trained on harder augmentations, like noise or rotation, were stronger and better at working with different kinds of astronomical images.

From Table 15, Table 16, Table 17, Table 18, Table 19 and Table 20 show the results that the Weighted Voting ensemble whose initial parameters are the values from Table 15, significantly enhances image classification across various augmented datasets—original, rotation, noise, HFlip, and VFlip. The model achieved its best performance on the original dataset (accuracy = 0.9926, F1 (real) = 0.9722), showing near-perfect classification. Performance decreased with rotation (accuracy = 0.7628, F1 = 0.6928) and dropped sharply with noise (accuracy = 0.5013, F1 = 0.0054), indicating difficulty handling noisy or rotated data. By contrast, HFlip and VFlip performed strongly (accuracy = 0.9703–0.9760, F1 = 0.9695–0.9763), proving the model’s ability to learn symmetrical spatial features. Overall, the Weighted Voting ensemble improved accuracy and stability for structured, symmetrical data but remained sensitive to noise and rotation, highlighting the need for better noise-handling and adaptive weighting to enhance real-world astronomical image classification.

From Table 21, Table 22, Table 23, Table 24, Table 25 and Table 26 show the results that the Weighted Voting ensemble (second ensemble) whose initial parameters are the values from Table 21 greatly improved performance, especially for the rotation and noise datasets, which were weaknesses in the previous experiment. Increasing the weight of Xception (fine-tuned and noise) from 0.3 to 0.5 boosted the noise dataset performance from accuracy = 0.50 to 0.6217 and F1 (real) = 0.0054 to 0.407, showing better noise handling. The rotation dataset also improved notably (accuracy = 0.9203, F1 = 0.915). Meanwhile, the original, HFlip, and VFlip datasets maintained strong results (accuracy > 0.97, F1 ≈ 0.97), confirming effective recognition of symmetrical spatial patterns. Overall, adjusting ensemble weights enhanced system stability and accuracy, particularly in classifying real astronomical objects under noisy or distorted conditions.

From Table 27, Table 28, Table 29, Table 30, Table 31 and Table 32 show the results that the Weighted Voting ensemble (third ensemble) whose initial parameters are the values from Table 27 achieved the best overall performance after increasing the weight of the Xception (fine-tuned and noise) model to 0.8, greatly enhancing noise robustness while maintaining strong performance on other augmentations. The original dataset reached accuracy = 0.9842 and F1 (real) = 0.9411, while the rotation dataset remained stable (accuracy = 0.9, F1 = 0.8965), reflecting improved balance between precision and recall. The most notable gain was in the noise dataset, which displayed accuracy = 0.885 and F1 (real) = 0.897, showing strong resistance to signal distortion. HFlip and VFlip also maintained excellent results (F1 > 0.96). Overall, increasing the weight of the noise-trained model significantly enhanced ensemble reliability and robustness, making it more effective for real-world astronomical image analysis under noisy and variable conditions.

### 3.7. Comparison with Previous Experimental Results

The present study builds upon the foundational research conducted by Tabacolde [56] and Liu et al. [57], who were innovators in utilizing transient images from the GOTO sky survey for real-bogus classification. Tabacolde et al. [56] first turned images into feature vectors and then used standard machine learning models to sort them. However, their method had many problems because there were many more false detections than real transients. This was because the classes were not balanced. Attempts were made to resolve this issue through oversampling and undersampling techniques; however, these methods had inherent limitations, including the risk of overfitting or the potential to exclude valuable data. Later, Liu et al. [57] proposed an alternative method to augment the sample size of the minority class by incorporating real images via rotations. This method did improve performance a little, but it still used traditional learning methods and could not directly obtain deep hierarchical features from raw image data. To overcome these limitations, the present study employs deep learning techniques, specifically Convolutional Neural Networks (CNNs), which can independently extract complex feature representations from raw images without manual intervention. CNNs can also deal with the complex spatial patterns that are common in pictures of the stars. This experiment uses a number of data augmentation methods, like rotation, flipping, and noise injection, to make the model better at generalizing to different types of data distortions. This study also uses transfer learning and fine-tuning to make the model work better. We also employ an ensemble learning strategy called Weighted Voting, which combines several high-performing CNN models and changes their predictions based on how well they do on validation. This method fixes the problems with each model and makes the classification system stronger, especially for images that are hard to read or have a lot of noise. This deep learning method using an ensemble is much more accurate and generalizes better than older methods when it comes to classifying real and bogus data. Table 33 represents the comparison with previous experimental results.

## 4. Discussion

The experimental results clearly demonstrate the effectiveness of integrating advanced Convolutional Neural Network (CNN) architectures with strategic data augmentation, transfer learning, fine-tuning, and ensemble learning in the context of astronomical transient detection. Several critical insights can be drawn from these findings.

### 4.1. Performance of Individual CNN Models

Across all augmentation strategies, models such as Xception, MobileNet, and VGG16/19 consistently outperformed others in both accuracy and F1-score. In particular, Xception with transfer learning and batch size 128 achieved the highest performance on rotation-augmented data, while MobileNet (Fine-Tuned) performed exceptionally on the original dataset. These results emphasize the flexibility and structural robustness of certain architectures in learning domain-specific patterns, especially those involving spatial symmetry, orientation variations, and brightness inconsistencies commonly observed in transient astronomical imagery.

However, not all architectures responded equally well to each augmentation technique. For instance, ResNet50 and ResNet101 displayed significant performance degradation under some configurations, most notably with noise-augmented data and larger batch sizes. This suggests that deeper architecture may require more sophisticated regularization or denoising techniques when handling corrupt or low-SNR images.

### 4.2. Impact of Data Augmentation

Among the augmentation strategies tested, horizontal flip (HFlip) and vertical flip (VFlip) yielded the most consistently high classification performance across all models, often resulting in F1-scores above 0.99. This indicates that these transformations effectively simulate the positional variability of transient objects and assist CNNs in learning rotational-invariant features.

By contrast, noise augmentation proved to be the most challenging for nearly all models. Most architectures failed completely under noisy conditions, yielding F1-scores as low as 0.000, indicating severe overfitting or class bias. Only a few models, notably Xception and ResNet50, managed to retain marginal classification ability. This highlights a key vulnerability in current deep learning models applied to astronomical imagery: their sensitivity to image noise, which is a prevalent issue in real observational data.

These findings emphasize the need for future research to focus on improving robustness against noise—either through preprocessing (e.g., denoising filters), adversarial training, or noise-aware model architectures.

### 4.3. Effectiveness of Ensemble Learning

To address the weaknesses of individual models—particularly under noise-augmented conditions—this study explored Soft Voting and Weighted Voting ensemble strategies. The Soft Voting ensemble offered moderate improvements in accuracy but remained limited by its uniform weighting scheme, which failed to sufficiently correct performance imbalance under noisy conditions. Specifically, recall in the “real” class was consistently low, indicating a failure to detect true transient events reliably.

By contrast, Weighted Voting facilitated more flexibility by assigning higher influence to noise-trained models. Progressive weight tuning in three ensemble configurations demonstrated that increasing the weight of the noise-robust model (up to 0.8) significantly improved overall performance, culminating in an F1-score (real) of 0.931 and an overall accuracy of 0.9348. This supports the hypothesis that ensemble strategies that explicitly compensate for weak conditions—such as noise corruption—can substantially improve model generalization and classification balance.

Notably, the final ensemble configuration not only corrected recall degradation but also maintained strong performance across other augmentation types, demonstrating its adaptability and robustness in complex, heterogeneous astronomical datasets.

### 4.4. Generalization and Scalability

The success of this ensemble approach illustrates the potential of combining multiple specialized models to form a unified system capable of handling the high variance and complexity of real-world astronomical data. The pipeline’s design—featuring modular training, augmentation-specific modeling, and intelligent voting—offers a scalable solution that can be extended to other sky surveys and transient detection projects. Furthermore, the system’s reliance on transfer learning and moderate fine-tuning suggests that this approach is computationally efficient and suitable for deployment in time-critical applications, such as real-time transient detection in large-scale surveys like GOTO.

### 4.5. Broader Implications for Astronomical Investigation and Data Analysis

Our findings hold broader significance for both astronomical investigation and data analysis in time-domain astronomy. The proposed ensemble deep learning framework enhances transient detection reliability by improving accuracy, reducing false positives, and increasing robustness against noise and image artifacts. These improvements enable faster and more confident identification of real astrophysical events such as supernovae, kilonovae, and variable stars, supporting timely follow-up observations and more efficient telescope resource allocation. From a data analysis standpoint, combining multiple pretrained CNN architectures provides a diverse representation of astronomical image features, effectively addressing issues such as class imbalance and PSF variation. This multi-model integration demonstrates strong data efficiency even under limited training samples. Furthermore, the framework offers scalability for applications across future large-scale surveys, such as LSST and Pan-STARRS, where automated, interpretable, and robust AI models are essential.

## 5. Conclusions

This study introduced an ensemble-based deep learning framework designed to classify real and bogus astronomical transients from sky survey images. By integrating transfer learning, fine-tuning, multiple data augmentation strategies (such as rotation, horizontal flip, vertical flip, and noise), and ensemble learning techniques, the proposed system achieved substantial improvements in classification accuracy and robustness. Models like Xception, MobileNet, and VGG19 consistently outperformed others, particularly under augmentation strategies that introduced geometric variations. While most models struggled with noise-injected images, the Weighted Voting strategy—especially when assigning higher weights to noise-trained models—greatly enhanced the system’s resilience to distortion and improved the F1-score of the “real” class to 0.931 with an overall accuracy of 93.48%. These results highlight the importance of model diversity and strategic ensemble configuration in addressing the challenges of real-world astronomical datasets. The proposed approach offers a scalable and practical solution for transient detection tasks in large-scale sky surveys and lays the groundwork for future research in noise-resilient deep learning for astronomy.

## 6. Future Work

In future work, we aim to extend the proposed ensemble deep learning framework toward a more multimodal, interpretable, and physically consistent system for astronomical transient detection. The current study demonstrates strong performance on simulated and GOTO datasets; however, several directions can improve both the scientific robustness and operational generalization. First, we plan to develop a lightweight visual–language integration framework that combines image-based features with textual metadata such as FITS headers, World Coordinate System (WCS) parameters, and observing conditions (e.g., sky brightness, exposure time, and telescope pointing). This multimodal pathway will allow the network to condition its predictions on contextual information and generate natural-language explanations for its decisions. Inspired by advances in vision–language modeling, such as “From Gaze to Insight”, this direction will enhance interpretability and pave the way for interactive AI-assisted discovery pipelines where astronomers can query model rationales in real time. Second, we intend to address the limitations caused by image upsampling from 21 × 21 to 224 × 224 pixels. While necessary for compatibility with ImageNet-pretrained architectures, this resizing may introduce interpolation artifacts and diminish the fidelity of point-spread function (PSF) structures. Future studies will explore resolution-aware and compact CNN architectures capable of operating directly on native resolutions, including anti-aliased convolutional stems, patchified feature extractors, and shallow networks optimized for small-scale astronomical images. Third, we plan to expand model validation to real GOTO difference images obtained from multiple observing nights and sky fields. These experiments will include genuine artifacts such as PSF distortions, ghost reflections, and saturation trails to evaluate the framework’s real-world robustness. Techniques such as domain adaptation and artifact-aware augmentation will also be explored to bridge the gap between simulated and observational data distributions. Finally, future investigations will assess the cross-survey applicability of the proposed approach under varying instrumental and environmental conditions. This includes benchmarking the ensemble framework using real data from other surveys (e.g., ZTF, LSST, and Pan-STARRS) and exploring its adaptability to related astrophysical tasks, such as variable star classification and optical counterpart identification in multi-messenger astronomy. Collectively, these future developments will advance the framework toward a scalable, interpretable, and physically reliable AI system capable of supporting next-generation time-domain surveys, enhancing both the automation and scientific value of transient detection in modern astronomy.

## Figures and Tables

**Figure 1 biomimetics-10-00781-f001:**
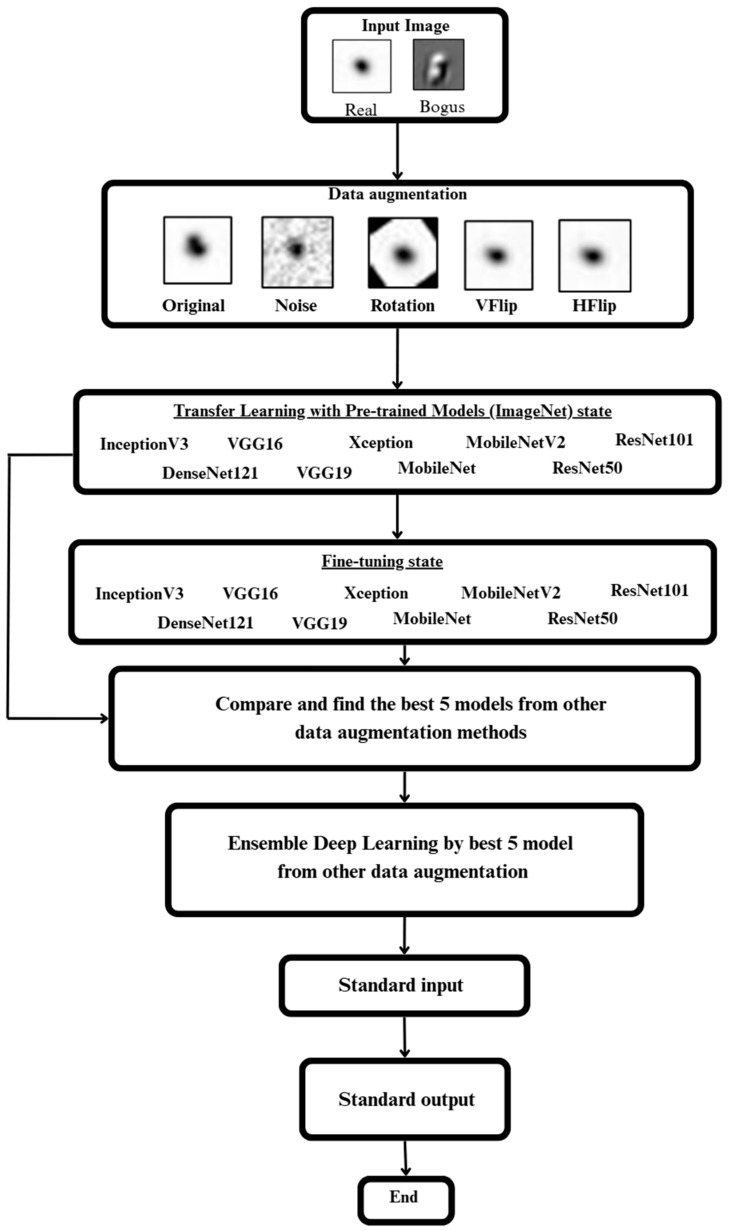
Figure depicting the overall methods used.

**Figure 2 biomimetics-10-00781-f002:**
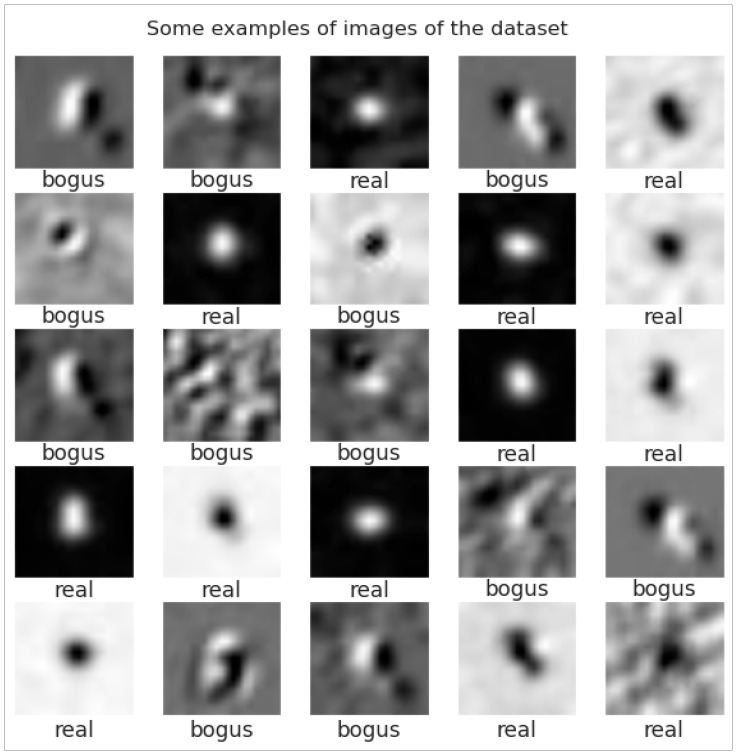
Some examples of images in the dataset.

**Figure 3 biomimetics-10-00781-f003:**
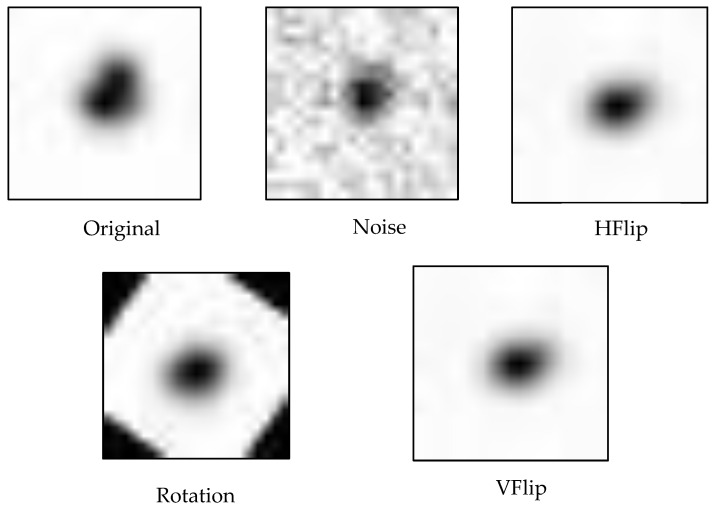
An example of data augmentation.

**Figure 4 biomimetics-10-00781-f004:**
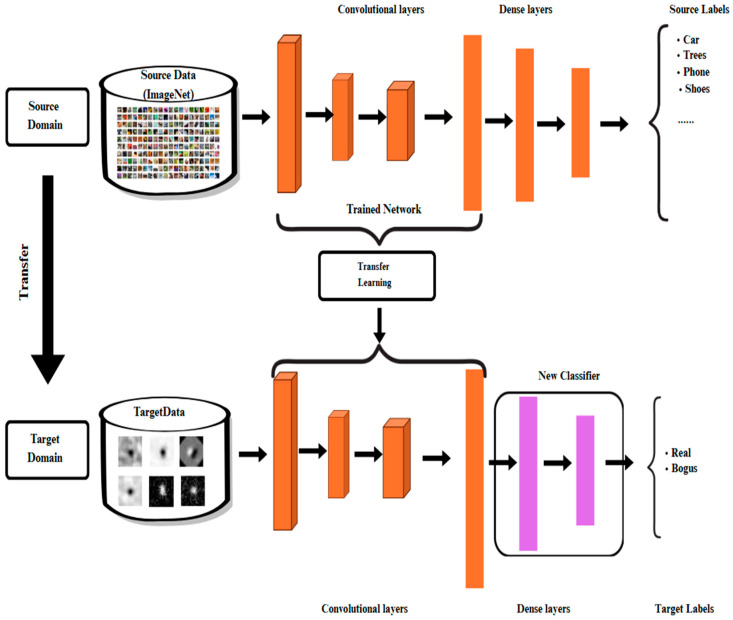
The process of transferring learned features from a source domain (ImageNet) to a target domain (astronomical transient data) using the transfer learning approach.

**Figure 5 biomimetics-10-00781-f005:**
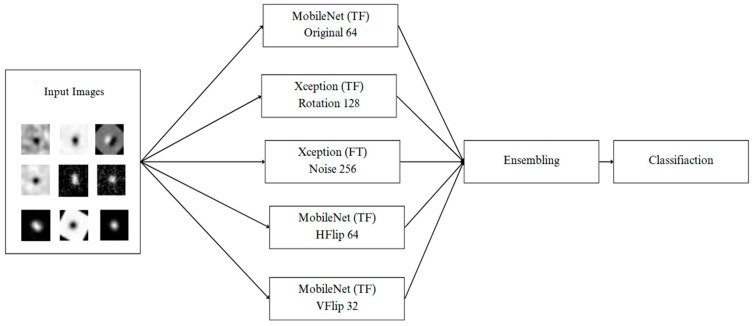
Ensemble architecture combining multiple CNN models trained under different augmentation strategies and batch sizes.

**Figure 6 biomimetics-10-00781-f006:**
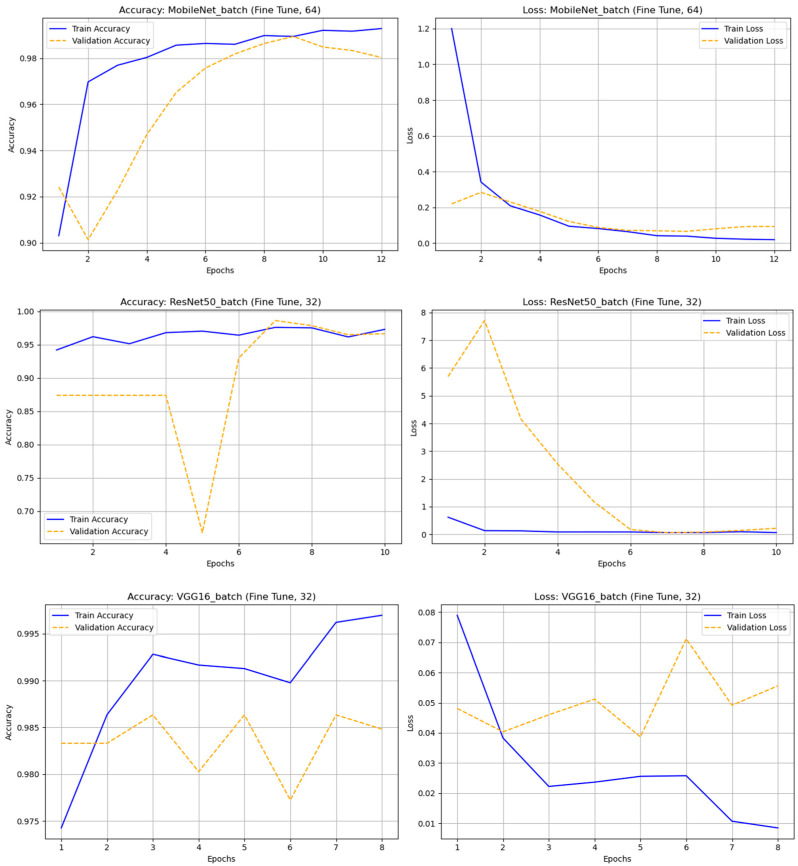
The accuracy and loss graph of different deep learning techniques with original (nonaugmented) dataset.

**Figure 7 biomimetics-10-00781-f007:**
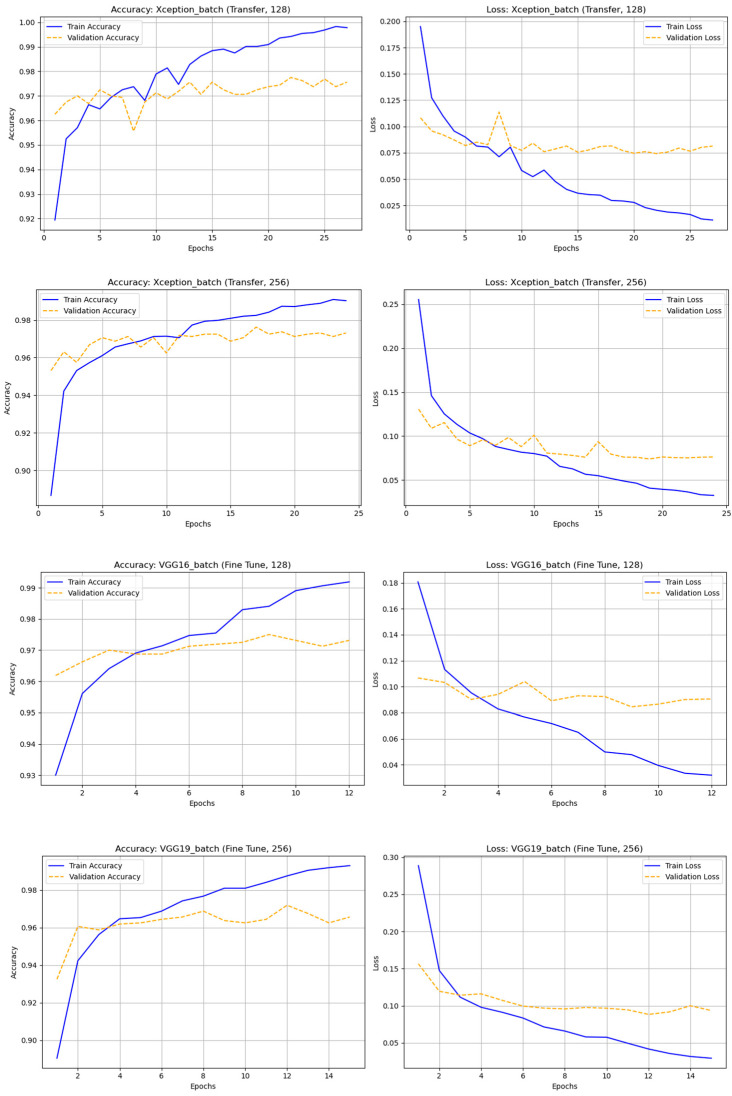
The accuracy and loss graph of different deep learning techniques with rotation dataset.

**Figure 8 biomimetics-10-00781-f008:**
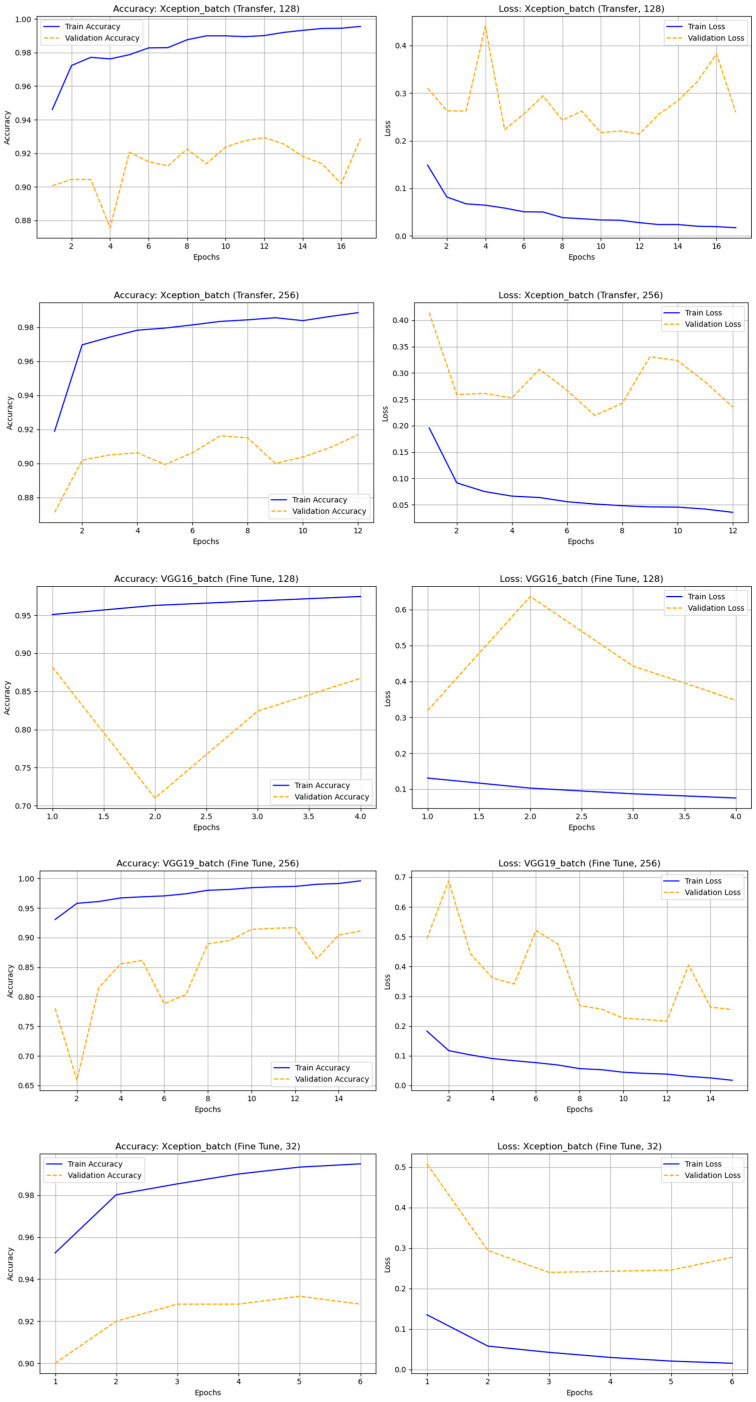
The accuracy and loss graph of different deep learning techniques with noise dataset.

**Figure 9 biomimetics-10-00781-f009:**
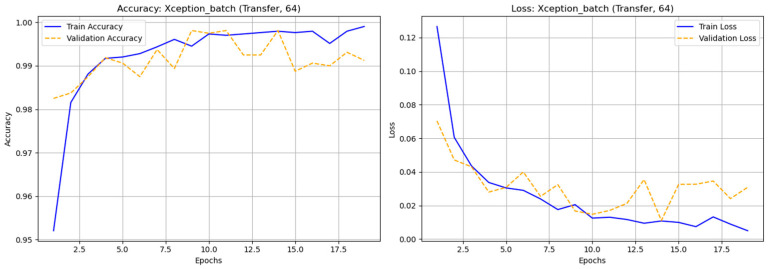
The accuracy and loss graph of different deep learning techniques with hflip dataset.

**Figure 10 biomimetics-10-00781-f010:**
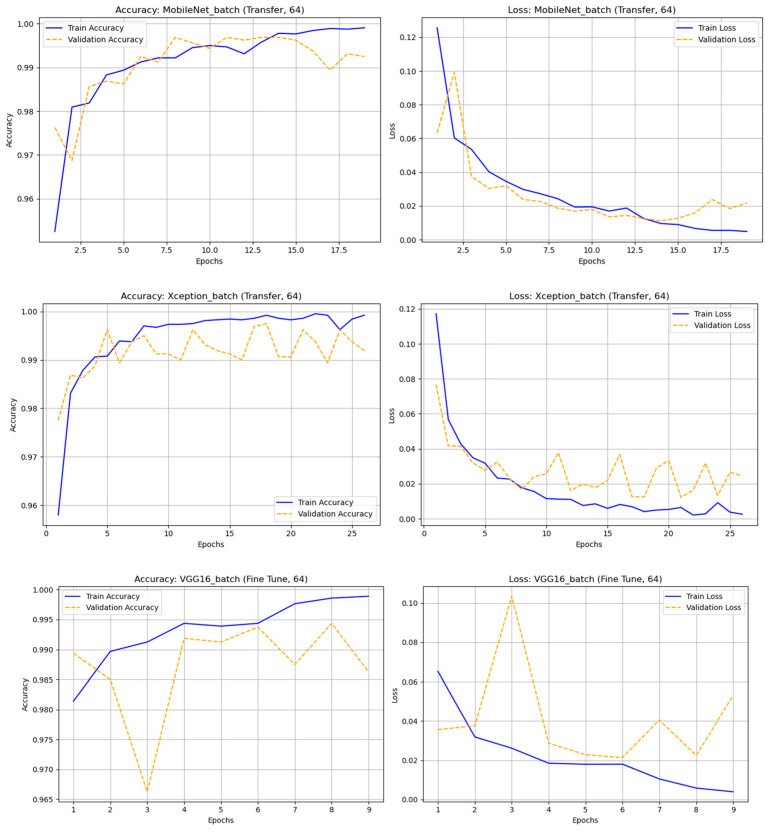
The accuracy and loss graph of different deep learning techniques with vflip dataset.

**Table 1 biomimetics-10-00781-t001:** Training data before and after oversampling, applied to address the class imbalance problem.

Training Data	Bogus	Real
Before Oversampling	2862	418
After Oversampling	4000	4000

**Table 2 biomimetics-10-00781-t002:** Hyperparameter settings used for training and fine-tuning pretrained convolutional and deep learning architectures (e.g., VGG, ResNet, DenseNet, MobileNet, Xception, and InceptionV3) for astronomical transient image classification.

Parameter	Value
Batch Size	3,264,128,256
Epoch	100, Early Stopping (Patience = 3)
Learning	0.001 (TF), 0.00001 (FT)
Optimizer	Adam
Loss Function	Binary Cross-Entropy
Fine-Tuning Unlocks	Top 30%

**Table 3 biomimetics-10-00781-t003:** Performance indicators and formula.

Performance Indicators	Formula
Precision (P)	$\frac{TruePositive}{TruePositive+FalsePositive}$
Recall (R)	$\frac{TruePositive}{TruePositive+FalseNagative}$
F1-score	$\frac{2*Positive*Recall}{Positive+Recall}$
Performance indicators	formula
Precision (P)	$\frac{TruePositive}{TruePositive+FalsePositive}$
Recall (R)	$\frac{TruePositive}{TruePositive+FalseNagative}$

**Table 4 biomimetics-10-00781-t004:** Comparison of classification results of different deep learning techniques with original dataset.

Rank	Model	Method	Batch Size	Accuracy	F1-Score (Bogus)	F1-Score (Real)
1	MobileNet	fine-tuned	64	0.98938	0.99393	0.95758
2	ResNet50	fine-tuned	32	0.98634	0.99222	0.94410
3	VGG16	fine-tuned	32	0.98634	0.99221	0.94479
4	VGG19	fine-tuned	256	0.98331	0.99049	0.93168
5	MobileNet	transfer	32	0.98483	0.99130	0.94048

**Table 5 biomimetics-10-00781-t005:** Comparison of classification results of different deep learning techniques with rotation dataset.

Rank	Model	Method	Batch Size	Accuracy	F1-Score (Bogus)	F1-Score (Real)
1	Xception	transfer	128	0.97750	0.97739	0.97761
2	Xception	transfer	256	0.97375	0.97352	0.97398
3	VGG16	fine-tuned	128	0.97500	0.97497	0.97503
4	VGG19	fine-tuned	256	0.97188	0.97168	0.97207
5	Xception	fine-tuned	32	0.97188	0.97146	0.97227

**Table 6 biomimetics-10-00781-t006:** Comparison of classification results of different deep learning techniques with noise dataset.

Rank	Model	Method	Batch Size	Accuracy	F1-Score (Bogus)	F1-Score (Real)
1	ResNet50	fine-tuned	256	0.85000	0.86014	0.83827
2	Xception	fine-tuned	256	0.72625	0.76776	0.6667
3	Xception	transfer	256	0.64000	0.72932	0.46269
4	Xception	fine-tuned	32	0.65500	0.74085	0.48411
5	InceptionV3	transfer	256	0.62125	0.72329	0.4000

**Table 7 biomimetics-10-00781-t007:** Comparison of classification results of different deep learning techniques with HFlip dataset.

Rank	Model	Method	Batch Size	Accuracy	F1-Score (Bogus)	F1-Score (Real)
1	MobileNet	transfer	64	0.99875	0.99875	0.99875
2	Xception	transfer	64	0.99875	0.99875	0.99875
3	VGG19	fine-tuned	64	0.99875	0.99875	0.99875
4	VGG16	fine-tuned	64	0.99687	0.99688	0.99688
5	MobileNetV2	transfer	256	0.99375	0.99379	0.99379

**Table 8 biomimetics-10-00781-t008:** Comparison of classification results of different deep learning techniques with VFlip dataset.

Rank	Model	Method	Batch Size	Accuracy	F1-Score (Bogus)	F1-Score (Real)
1	MobileNet	transfer	32	0.99875	0.99875	0.99875
2	VGG19	fine-tuned	32	0.99875	0.99875	0.99875
3	InceptionV3	transfer	32	0.99750	0.99749	0.99751
4	MobileNet	fine-tuned	32	0.99750	0.99749	0.99751
5	Xception	transfer	32	0.99687	0.99687	0.99688

**Table 9 biomimetics-10-00781-t009:** Selected models for ensemble learning based on best performance by augmentation type.

Model	Method	Augmentation	Batch Size
MobileNet	Fine-Tuned	Original	64
Xception	Transfer	Rotation	128
Xception	Fine-Tuned	Noise	256
MobileNet	Transfer	HFlip	64
MobileNet	Transfer	VFlip	32

**Table 10 biomimetics-10-00781-t010:** Test with original data.

Model	Accuracy	Precision (Bogus)	Recall (Bogus)	F1-Score (Bogus)	Precision (Real)	Recall (Real)	F1-Score (Real)
Ensemble	0.9937	0.9986	0.9930	0.9958	0.9541	0.9905	0.9720
Confusion Matrix							
		Pred: bogus			Pred: real		
True: bogus		711			5		
True: real		1			104		

**Table 11 biomimetics-10-00781-t011:** Test with rotation data.

Model	Accuracy	Precision (Bogus)	Recall (Bogus)	F1-Score (Bogus)	Precision (Real)	Recall (Real)	F1-Score (Real)
Ensemble	0.7386	0.6584	0.992	0.7914	0.984	0.4852	0.6499
Confusion Matrix							
		Pred: bogus			Pred: real		
True: bogus		3968			32		
True: real		2059			1941		

**Table 12 biomimetics-10-00781-t012:** Test with noise data.

Model	Accuracy	Precision (Bogus)	Recall (Bogus)	F1-Score (Bogus)	Precision (Real)	Recall (Real)	F1-Score (Real)
Ensemble	0.5006	0.5	1	0.6667	1	0.0012	0.0024
Confusion Matrix							
		Pred: bogus			Pred: real		
True: bogus		4000			0		
True: real		3995			5		

**Table 13 biomimetics-10-00781-t013:** Test with HFlip data.

Model	Accuracy	Precision (Bogus)	Recall (Bogus)	F1-Score (Bogus)	Precision (Real)	Recall (Real)	F1-Score (Real)
Ensemble	0.9667	0.946	0.99	0.967	0.9895	0.9435	0.966
Confusion Matrix							
		Pred: bogus			Pred: real		
True: bogus		3960			40		
True: real		226			3774		

**Table 14 biomimetics-10-00781-t014:** Test with VFlip data.

Model	Accuracy	Precision (Bogus)	Recall (Bogus)	F1-Score (Bogus)	Precision (Real)	Recall (Real)	F1-Score (Real)
Ensemble	0.9889	0.9848	0.993	0.989	0.9929	0.9847	0.9888
Confusion Matrix							
		Pred: bogus			Pred: real		
True: bogus		3972			28		
True: real		61			3939		

**Table 15 biomimetics-10-00781-t015:** Parameters in first ensemble.

Model	Method	Batch Size	Augmentation	Weight
MobileNet	fine_tuned	64	Original	0.2
Xception	transfer	128	Rotation	0.2
Xception	fine_tuned	256	Noise	0.3
MobileNet	transfer	64	HFlip	0.15
MobileNet	Transfer	32	VFlip	0.15

**Table 16 biomimetics-10-00781-t016:** Test with original data.

Model	Accuracy	Precision (Bogus)	Recall (Bogus)	F1-Score (Bogus)	Precision (Real)	Recall (Real)	F1-Score (Real)
Ensemble	0.9926	1	0.9916	0.9958	0.9459	1	0.9722
Confusion Matrix							
		Pred: bogus			Pred: real		
True: bogus		710			6		
True: real		0			105		

**Table 17 biomimetics-10-00781-t017:** Test with rotation data.

Model	Accuracy	Precision (Bogus)	Recall (Bogus)	F1-Score (Bogus)	Precision (Real)	Recall (Real)	F1-Score (Real)
Ensemble	0.7628	0.6805	0.9907	0.8068	0.983	0.535	0.6928
Confusion Matrix							
		Pred: bogus			Pred: real		
True: bogus		3963			37		
True: real		1560			2140		

**Table 18 biomimetics-10-00781-t018:** Test with noise data.

Model	Accuracy	Precision (Bogus)	Recall (Bogus)	F1-Score (Bogus)	Precision (Real)	Recall (Real)	F1-Score (Real)
Ensemble	0.5013	0.5	1	0.6672	1	0.0027	0.0054
Confusion Matrix							
		Pred: bogus			Pred: real		
True: bogus		4000			0		
True: real		3959			11		

**Table 19 biomimetics-10-00781-t019:** Test with HFlip data.

Model	Accuracy	Precision (Bogus)	Recall (Bogus)	F1-Score (Bogus)	Precision (Real)	Recall (Real)	F1-Score (Real)
Ensemble	0.9703	0.9556	0.9865	0.9708	0.986	0.9542	0.9698
Confusion Matrix							
		Pred: bogus			Pred: real		
True: bogus		3946			54		
True: real		183			3817		

**Table 20 biomimetics-10-00781-t020:** Test with VFlip data.

Model	Accuracy	Precision (Bogus)	Recall (Bogus)	F1-Score (Bogus)	Precision (Real)	Recall (Real)	F1-Score (Real)
Ensemble	0.976	0.9625	0.9905	0.9763	0.9902	0.9615	0.9756
Confusion Matrix							
		Pred: bogus			Pred: real		
True: bogus		3962			38		
True: real		154			3846		

**Table 21 biomimetics-10-00781-t021:** Parameters in second ensemble.

Model	Method	Batch Size	Augmentation	Weight
MobileNet	fine_tuned	64	Original	0.2
Xception	transfer	128	Rotation	0.2
Xception	fine_tuned	256	Noise	0.50
MobileNet	transfer	64	HFlip	0.15
MobileNet	transfer	32	VFlip	0.15

**Table 22 biomimetics-10-00781-t022:** Test with original data.

Model	Accuracy	Precision (Bogus)	Recall (Bogus)	F1-Score (Bogus)	Precision (Real)	Recall (Real)	F1-Score (Real)
Ensemble	0.9902	1	0.9888	0.9943	0.9292	1	0.9633
Confusion Matrix							
		Pred: bogus			Pred: real		
True: bogus		708			8		
True: real		0			105		

**Table 23 biomimetics-10-00781-t023:** Test with rotation data.

Model	Accuracy	Precision (Bogus)	Recall (Bogus)	F1-Score (Bogus)	Precision (Real)	Recall (Real)	F1-Score (Real)
Ensemble	0.9203	0.8734	0.98325	0.9250	0.9808	0.8575	0.915
Confusion Matrix							
		Pred: bogus			Pred: real		
True: bogus		3933			67		
True: real		570			3430		

**Table 24 biomimetics-10-00781-t024:** Test with noise data.

Model	Accuracy	Precision (Bogus)	Recall (Bogus)	F1-Score (Bogus)	Precision (Real)	Recall (Real)	F1-Score (Real)
Ensemble	0.6217	0.5706	0.9832	0.722	0.9395	0.2602	0.407
Confusion Matrix							
		Pred: bogus			Pred: real		
True: bogus		3933			67		
True: real		2959			1041		

**Table 25 biomimetics-10-00781-t025:** Test with HFlip data.

Model	Accuracy	Precision (Bogus)	Recall (Bogus)	F1-Score (Bogus)	Precision (Real)	Recall (Real)	F1-Score (Real)
Ensemble	0.97125	0.9631	0.98	0.9714	0.9796	0.9625	0.970
Confusion Matrix							
		Pred: bogus			Pred: real		
True: bogus		3920			80		
True: real		150			3850		

**Table 26 biomimetics-10-00781-t026:** Test with VFlip data.

Model	Accuracy	Precision (Bogus)	Recall (Bogus)	F1-Score (Bogus)	Precision (Real)	Recall (Real)	F1-Score (Real)
Ensemble	0.971	0.9584	0.9847	0.9714	0.9843	0.9572	0.9706
Confusion Matrix							
		Pred: bogus			Pred: real		
True: bogus		3939			61		
True: real		171			3829		

**Table 27 biomimetics-10-00781-t027:** Parameters in third ensemble.

Model	Method	Batch Size	Augmentation	Weight
MobileNet	fine_tuned	64	Original	0.20
Xception	transfer	128	Rotation	0.20
Xception	fine_tuned	256	Noise	0.80
MobileNet	transfer	64	HFlip	0.15
MobileNet	transfer	32	VFlip	0.15

**Table 28 biomimetics-10-00781-t028:** Test with original data.

Model	Accuracy	Precision (Bogus)	Recall (Bogus)	F1-Score (Bogus)	Precision (Real)	Recall (Real)	F1-Score (Real)
Ensemble	0.9842	0.9986	0.9832	0.991	0.8966	0.9904	0.9411
Confusion Matrix							
		Pred: bogus			Pred: real		
True: bogus		704			12		
True: real		1			104		

**Table 29 biomimetics-10-00781-t029:** Test with rotation data.

Model	Accuracy	Precision (Bogus)	Recall (Bogus)	F1-Score (Bogus)	Precision (Real)	Recall (Real)	F1-Score (Real)
Ensemble	0.9	0.872	0.9377	0.9	0.9327	0.863	0.8965
Confusion Matrix							
		Pred: bogus			Pred: real		
True: bogus		3751			249		
True: real		548			3452		

**Table 30 biomimetics-10-00781-t030:** Test with noise data.

Model	Accuracy	Precision (Bogus)	Recall (Bogus)	F1-Score (Bogus)	Precision (Real)	Recall (Real)	F1-Score (Real)
Ensemble	0.885	0.9993	0.771	0.870	0.813	0.9995	0.897
Confusion Matrix							
		Pred: bogus			Pred: real		
True: bogus		3084			916		
True: real		2			3998		

**Table 31 biomimetics-10-00781-t031:** Test with HFlip data.

Model	Accuracy	Precision (Bogus)	Recall (Bogus)	F1-Score (Bogus)	Precision (Real)	Recall (Real)	F1-Score (Real)
Ensemble	0.9686	0.9694	0.9686	0.9678	0.9678	0.9695	0.9686
Confusion Matrix							
		Pred: bogus			Pred: real		
True: bogus		3871			129		
True: real		122			3878		

**Table 32 biomimetics-10-00781-t032:** Test with VFlip data.

Model	Accuracy	Precision (Bogus)	Recall (Bogus)	F1-Score (Bogus)	Precision (Real)	Recall (Real)	F1-Score (Real)
Ensemble	0.9617	0.9545	0.9692	0.962	0.968	0.9542	0.9614
Confusion Matrix							
		Pred: bogus			Pred: real		
True: bogus		3877			123		
True: real		183			3817		

**Table 33 biomimetics-10-00781-t033:** Comparison with previous experimental results.

Ref./Year	Technique	Dataset	Accuracy/F1 (Class 1)
Tabacolde et al. [56]	ML with handcrafted features (e.g., SVM, Decision Tree)	GOTO	RF best precision, but recall < 0.1
Liu et al. [57]	CNN baseline (1 conv layer) with multiple optimizers and augmentation	GOTO	F1-class 1 = 0.917 (best at batch size 128)
Proposed	Ensemble of fine-tuned CNNs with Weighted Voting with data augmentation	GOTO	F1-class 1 = 0.9717

## Data Availability

The original contributions presented in this study are included in the article. Further inquiries can be directed to the corresponding authors.

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
