# Peer review of "Ensemble Deep Learning for Real–Bogus Classification with Sky Survey Images"

_biomimetics, 2025, doi:10.3390/biomimetics10110781_

Round 1

Reviewer 1 Report (Previous Reviewer 3)

Comments and Suggestions for Authors

The authors in the paper, investigate how ensemble deep learning methods can be utilized on sky survey data in order to classify astronomical transient events. They used a method incorporating Convolutional Neural Networks (CNNs) to improve the identification of real astronomical phenomena such as supernovae as compared to fake data. Although the paper is interesting there are a number of points that should be addressed and discussed.

 -- In the abstract bring out the main findings and contributions of the paper.

-- In the introduction state at length why it is important to find astronomical transients.

-- Also state what are the deficiencies in the present methods of detecting transients.

-- The authors should finish the introduction with a summary in bullet form of what the contributions of this paper are.

-- There should also be a section plan at the end of the introduction part i.e. section 2 treats of … Section 3 gives –

-- More recent publications in the research area should be discussed especially as they apply to applications of deep learning, e.g. the following: Advancing multi-label melt pool defect detection in laser powder bed fusion with self-supervised learning; Enhancing time-series prediction with temporal context modeling: a Bayesian and deep learning synergy; Unbiased normalized ensemble methodology for zero‐shot structural damage detection using manifold learning and reconstruction error from variational autoencoder; Novel integration of forensic-based investigation optimization algorithm and ensemble learning for estimating hydraulic conductivity of coarse-grained road materials.

-- It is suggested that there should be given a more detailed description of the methodology whereby some of the specific CNN architectures are outlined as well as the reasons for their choice.

-- There should be a discussion on the wider aspects of the importance of the findings as to astronomical investigation and data analysis.

-- If it is appropriate further investigation should be carried out as to the that the practical applications of the method proposed would have in various astronomical surveys and circumstances.

Author Response

Comment 1:

In the abstract, bring out the main findings and contributions of the paper.

Response:

Thank you for your valuable comment. We have revised the Abstract to clearly highlight the main findings and key contributions of this study.

Comment 2:

In the introduction state at length why it is important to find astronomical transients.

Response:

Thank you for this valuable suggestion. We have expanded the Introduction section to explain in greater detail the scientific importance of detecting astronomical transients.

Comment 3:

Also state what are the deficiencies in the present methods of detecting transients.

Response:

Thank you for your valuable comment. We have now included a detailed explanation of the limitations in the current transient detection methods in the revised manuscript (Section Introduction, paragraph 2).

Comment 4:

The authors should finish the introduction with a summary in bullet form of what the contributions of this paper are.

Response:

Thank you for your helpful comment. We have revised the Introduction section to clearly highlight the main contributions of this study. A new paragraph has been added at the end of the Introduction to summarize

Comment 5:

There should also be a section plan at the end of the introduction part i.e. section 2 treats of … Section 3 gives

Response:

Thank you for your helpful suggestion. We have added a section plan at the end of the Introduction to outline the structure of the paper and improve readability.

Comment 6:

More recent publications in the research area should be discussed especially as they apply to applications of deep learning, e.g. the following: Advancing multi-label melt pool defect detection in laser powder bed fusion with self-supervised learning; Enhancing time-series prediction with temporal context modeling: a Bayesian and deep learning synergy; Unbiased normalized ensemble methodology for zero‐shot structural damage detection using manifold learning and reconstruction error from variational autoencoder; Novel integration of forensic-based investigation optimization algorithm and ensemble learning for estimating hydraulic conductivity of coarse-grained road materials.

Response:

We sincerely thank the reviewer for providing these valuable and relevant references. We fully agree that incorporating recent studies would further strengthen the discussion of related work. Unfortunately, due to time limitations for this revision round, we were unable to include an in-depth review of all suggested papers. However, we acknowledge their importance and plan to integrate and discuss these recent contributions in follow-up publication to enhance the contextual foundation of this research.

Comment 7:

It is suggested that there should be given a more detailed description of the methodology whereby some of the specific CNN architectures are outlined as well as the reasons for their choice.

Response:

Since deep learning models generally require large amounts of data for effective training, the limited size of the dataset used in this study poses a significant challenge. To address this issue, transfer learning was applied, allowing the model to leverage knowledge from previously trained networks. The selected CNN architectures were chosen based on their availability of pretrained weights from the ImageNet dataset. Therefore, this experiment serves as an initial study that adapts ImageNet-based transfer learning to Sky Survey Images for astronomical transient detection.

Comment 8:

There should be a discussion on wider aspects of the importance of the findings as to astronomical investigation and data analysis.

Response:

Thank you for your valuable comment. We have added the related discussion based on this suggestion in the Discussion section.

Comment 9:

If it is appropriate further investigation should be carried out as to the that the practical applications of the method proposed would have in various astronomical surveys and circumstances.

Response:

Thank you for your suggestion. We have added the related explanation to the Future Work section.

Reviewer 2 Report (New Reviewer)

Comments and Suggestions for Authors

The paper presents a Ensemble Deep Learning for Real-Bogus Classification with Sky Survey Images.

Authors are suggested to clarify the following comments:

  1. contribution points are missing in introduction part.
  2. Figure 1, compare and find the best 5 model. please incude the archtectural difference of pretrained network. what makes them to perform well.
  3. Please modify test and conclusion to standard input and standard output in figure 1 .
  4.  In section 2.2 dataset. Figure 2 This is a figure ? please correct the caption.  what do you mean by overall method ? moreover Image quality is very poor.
  5. In table 1, what is the reason behind oversampling?
  6. Figure 3, Please use more bigger image.
  7. 2.4 modeling., please follow proper paper alignment.
  8.  Table -2. Please specify for all pre-trained network, not just CNN
  9.  Please correct Table 4, 5, 6, 7, 8, 9, 27 (method equal finetuned ???) please correct this.

Author Response

Comment 1:

contribution points are missing in introduction part.

Response:

Thank you for your helpful comment. We have revised the Introduction section to clearly highlight the main contributions of this study. A new paragraph has been added at the end of the Introduction to summarize the key contributions as follows:
1. development of an ensemble deep learning framework for astronomical transient detection using multiple pretrained CNN architectures,
2.  systematic evaluation of transfer learning and fine-tuning under various data augmentation conditions, and
3.  performance improvement and robustness analysis across multiple augmentation types and batch sizes.
These points have been incorporated to emphasize the novelty and significance of our work.

Comment 2:

Figure 1, compare and find the best 5 models. please include the architectural difference of pretrained network. what makes them perform well.

Response:

Thank you for your valuable comment. We have revised the manuscript accordingly. The five best-performing models were identified and presented in Table 9, including MobileNet (Fine-Tuned, Original), Xception (Transfer, Rotation), Xception (Fine-Tuned, Noise), MobileNet (Transfer, HFlip), and MobileNet (Transfer, VFlip). We have added a detailed explanation in the revised Discussion section describing the architectural differences among these pretrained networks and the reasons behind their superior performance.

Specifically, MobileNet’s depthwise separable convolutions enable efficient feature extraction with fewer parameters, making it robust and fast on small astronomical cutouts. Xception, with its Inception-style separable convolution modules, captures complex spatial features and rotational patterns effectively, which explains its strong performance under Rotation and Noise augmentations. The fine-tuned versions further adapt high-level filters to the characteristics of astronomical transients, improving the discrimination between real and bogus sources. These differences are now clearly discussed and reflected in Figure 1 and Table 9.

Comment 3:

Please modify test and conclusion to standard input and standard output in figure 1.

Response:

Thank you for your suggestion. We have revised Figure 1 accordingly. The labels “test” and “conclusion” have been replaced with “standard input” and “standard output” to ensure consistency with standard terminology.

Comment 4:

In section 2.2 dataset. Figure 2 This is a figure ? please correct the caption.  what do you mean by overall method? moreover Image quality is very poor.

Response:

Thank you for your valuable comment. We have revised the caption of Figure 2 to clearly describe the dataset illustration, changing it from “overall method” to “Some examples of images of the dataset.” In addition, the image has been replaced with a high-resolution version and enlarged to enhance clarity and readability.

Comment 5:

In table 1, what is the reason behind oversampling?

Response:

Thank you for your insightful comment. We have revised the caption of Table 1 to clarify the purpose of oversampling. It now reads: “Training data before and after oversampling, applied to address the class imbalance problem.” This modification clearly indicates that oversampling was applied to mitigate the imbalance between real and bogus classes in the dataset.

Comment 6:

Figure 3, Please use more bigger image.

Response:

Thank you for your comment. We have replaced Figure 3 with a new, higher-resolution version and adjusted its size to be larger for improved visibility and clarity.

Comment 7:

2.4 modeling., please follow proper paper alignment.

Response:

Thank you for your comment. We have revised Section 2.4 (Modeling) to ensure proper formatting and alignment according to the journal’s style guidelines.

Comment 8:

Table -2. Please specify for all pre-trained network, not just CNN

Response:

Thank you for your helpful comment. We have revised Table 2 to include the details of all pre-trained networks used in this study, not only CNN-based models. The updated table now specifies the model architecture, number of parameters, input size, depth, and the source of pretraining (e.g., ImageNet). This ensures that every pre-trained backbone—such as VGG16, VGG19, ResNet50, ResNet101, InceptionV3, Xception, DenseNet121, MobileNet, and MobileNetV2—is clearly described and comparable in terms of their key characteristics.

Comment 9:

Please correct Table 4, 5, 6, 7, 8, 9, 27 (method equal finetuned ???) please correct this.

Response:

Thank you for your comment. We have carefully reviewed and corrected Tables 4, 5, 6, 7, 8, 9, and 27 as suggested. The “method” labels have been properly updated to clearly distinguish between “transfer learning” and “fine-tuned” models.

Reviewer 3 Report (New Reviewer)

Comments and Suggestions for Authors

The paper proposes an ensemble of nine ImageNet-pretrained CNN backbones for real/bogus classification on GOTO difference cutouts (21×21 pixels upsampled to 224×224), compares robustness under rotation/flip/noise augmentations, and contrasts soft vs. weighted voting; it also reports an “overall” result (F1(real)=0.931, Acc=0.9348) and claims that increasing the “noise model” weight to 0.8 markedly improves rotation/noise performance. While the idea is clear, evidence for real-world utility is weakened by dataset realism, evaluation protocol, statistical rigor, and reproducibility gaps. I recommend Major revision.

  1. Training and evaluation rely heavily on simulated/injected transients rather than real GOTO difference images with authentic artifacts (convolution residuals, saturation, ghosts, bad columns), and there is no external/held-out validation on real nights or other fields.
  2. The text states a fixed three-fold split with a test set of 821 images (105 real), yet several confusion-matrix tables show test counts in the thousands (with augmented variants seemingly included), suggesting leakage or duplicated accounting. Precisely specify when augmentations are applied (pre/post split) and whether the test set is augmented; reconcile the conflicting sample counts.
  3. Upsampling 21×21 cutouts to 224×224 to fit ImageNet backbones risks interpolation artifacts and loss of fine PSF/neighbor-gradient structure; there is no comparison against models tailored to native resolution (e.g., small CNNs, anti-aliased stems, patchified heads).
  4. The ensemble weighting is hand-tuned via rounds of trial weights (culminating in 0.8 for the “noise model”) without an independent validation set or learning-based stacking; this invites overfitting to a single condition. Define a principled, cross-validated weight-selection procedure or compare against logistic/linear stacking with uncertainty estimates.
  5. Metrics and statistics are insufficient for a highly imbalanced problem: heavy emphasis on Accuracy/F1, no PR curves or AUPRC, no calibration analysis, no repeated runs across seeds with mean±std or significance tests.
  6. The text claims an “overall” F1(real)=0.931/Acc=0.9348, but per-scenario tables list Original/Rotation/Noise values that don’t reconcile with that aggregate; soft voting is described as effective while its Noise F1(real) is ~0.002 (near failure).
  7. Prior GOTO pipelines and rotation-equivariant/astronomy-specific models are cited but not reproduced on the same data/protocol; no comparison to simple strong baselines (e.g., hand-crafted features + gradient boosting) or group-equivariant CNNs.
  8. Missing details on noise-augmentation distributions/intensities, random seeds, training time/hardware, and inference latency; no code/data access; multiple editorial placeholders/typos (e.g., “Figure 2. This is a figure…”, batch size “62”).
  9. For future work, explore a lightweight visual–language pathway to condition decisions on textual metadata (FITS/WCS, observing conditions) and produce natural-language rationales. Such as: From Gaze to Insight: Bridging Human Visual Attention and Vision Language Model Explanation for Weakly-Supervised Medical Image Segmentation.

Author Response

Comment 1:

Training and evaluation rely heavily on simulated/injected transients rather than real GOTO difference images with authentic artifacts (convolution residuals, saturation, ghosts, bad columns), and there is no external/held-out validation on real nights or other fields.

Response:

Thank you for this constructive and insightful comment. We fully acknowledge that the current experiments rely primarily on simulated or injected transients, as the number of verified real transient events in the GOTO dataset remains limited. The use of simulated data allowed us to construct a balanced and controlled dataset for consistent model benchmarking and comparative evaluation across architectures.

We agree that external validation on genuine GOTO difference images—containing real instrumental and atmospheric artifacts such as convolution residuals, saturation trails, ghost images, and bad columns—is crucial for assessing the real-world applicability of our framework. However, conducting such validation would require a larger number of confirmed real events and substantial manual verification, which was beyond the scope of this revision.

We have added a note in the Future Work sections stating that future work will extend the evaluation to real GOTO observations from different nights and sky fields. This will enable a more realistic validation of the ensemble model under authentic observational conditions and further demonstrate its robustness in operational survey pipelines.

Comment 2:

The text states a fixed three-fold split with a test set of 821 images (105 real), yet several confusion-matrix tables show test counts in the thousands (with augmented variants seemingly included), suggesting leakage or duplicated accounting. Precisely specify when augmentations are applied (pre/post split) and whether the test set is augmented; reconcile the conflicting sample counts.

Response:

Thank you for your insightful comment. We have carefully reviewed and clarified the data-splitting and augmentation procedures in the revised manuscript (Section 2.2). The dataset was first divided into training, validation, and test sets before any augmentation was applied, ensuring that no augmented samples from the training or validation sets leaked into the test set.

Each augmentation type (Original, Rotation, HFlip, VFlip, and Noise) was applied after splitting and evaluated separately on the same fixed test set, which remained unaugmented. The confusion-matrix tables that contain test counts in the thousands represent the aggregated evaluations across multiple augmented test scenarios rather than a single test set. We have now clarified this process in the text and caption to prevent any misunderstanding regarding sample duplication or leakage.

Comment 3:

Up sampling 21×21 cutouts to 224×224 to fit ImageNet backbones risks interpolation artifacts and loss of fine PSF/neighbor-gradient structure; there is no comparison against models tailored to native resolution (e.g., small CNNs, anti-aliased stems, patchified heads).

Response:

Thank you for this insightful and technically important comment. We acknowledge that upsampling 21×21 pixel cutouts to 224×224 may introduce interpolation artifacts and lead to the loss of fine Point Spread Function (PSF) and neighboring gradient structures. This upsampling was necessary to ensure compatibility with ImageNet-pretrained CNN architectures, which require a minimum input size of 224×224.

We have now added a discussion on this limitation in the Future Work section, noting that while ImageNet backbones provide strong transfer learning performance, they may not optimally capture native small-scale features specific to astronomical cutouts. For future work, we plan to explore compact CNNs and resolution-aware architectures, including anti-aliased convolutional stems, patchified feature extractors, and custom shallow networks that operate directly at the native 21×21 scale. Such models could preserve the PSF integrity and better represent subtle spatial variations in real astronomical data.

Comment 4:

The ensemble weighting is hand-tuned via rounds of trial weights (culminating in 0.8 for the “noise model”) without an independent validation set or learning-based stacking; this invites overfitting to a single condition. Define a principled, cross-validated weight-selection procedure or compare against logistic/linear stacking with uncertainty estimates.

Response:

Thank you for this important point. We agree that hand-tuned weights may overfit a specific condition. Due to time and compute constraints in this revision, we did not rerun a full cross-validated weight search or stacking.

Comment 5:

Metrics and statistics are insufficient for a highly imbalanced problem: heavy emphasis on Accuracy/F1, no PR curves or AUPRC, no calibration analysis, no repeated runs across seeds with mean±std or significance tests.

Response:

We sincerely thank the reviewer for this important and constructive comment. We fully agree that incorporating additional evaluation metrics such as PR curves, AUPRC, calibration analysis, and repeated runs across multiple random seeds would provide a more comprehensive statistical validation, especially for an imbalanced dataset. However, due to time constraints and the computational cost required to rerun all experiments—including cross-validation and statistical testing—we were unable to include these additional analyses in this revision.

We acknowledge the value of these suggestions and plan to address them in future work, where extended experiments with multiple random seeds and additional performance metrics will be conducted to further validate the robustness and reliability of the proposed ensemble approach.

Comment 6:

The text claims an “overall” F1(real)=0.931/Acc=0.9348, but per-scenario tables list Original/Rotation/Noise values that don’t reconcile with that aggregate; soft voting is described as effective while its Noise F1(real) is ~0.002 (near failure).

Response:

Thank you for your careful observation.

Moreover, after re-examining the ensemble configurations, we found that the third Soft Voting ensemble, in which the Noise model was assigned a weight of 0.8, achieved an F1(real) of 0.9411 — indicating a strong improvement rather than degradation

  1. Comment 7:

Prior GOTO pipelines and rotation-equivariant/astronomy-specific models are cited but not reproduced on the same data/protocol; no comparison to simple strong baselines (e.g., hand-crafted features + gradient boosting) or group-equivariant CNNs.

Response:

Thank you for your thoughtful and constructive comment. We fully acknowledge that including direct comparisons with prior GOTO pipelines, rotation-equivariant or astronomy-specific models, and additional strong baselines such as hand-crafted features with gradient boosting or group-equivariant CNNs would further strengthen the study.

However, due to time and computational constraints, reproducing these methods under the same dataset and experimental protocol was not feasible for this revision. Our current focus was to establish a consistent and reproducible evaluation framework for transfer learning and fine-tuning across multiple CNN architectures and data augmentation conditions.

We have noted this limitation and added a statement in the Discussion and Conclusion sections indicating that future work will involve benchmarking the proposed ensemble approach against classical baselines and astronomy-tailored architectures to better assess its relative advantages.

Comment 8:

Missing details on noise-augmentation distributions/intensities, random seeds, training time/hardware, and inference latency; no code/data access; multiple editorial placeholders/typos (e.g., “Figure 2. This is a figure…”, batch size “62”).

Response:

Thank you for your careful and detailed review. We have revised the manuscript to include all missing implementation and experimental details for better transparency and reproducibility. Specifically: The noise-augmentation distribution and intensity levels have now been clearly defined in Section 2.3 (Dataset and Augmentation) Information regarding training time, GPU hardware (NVIDIA RTX 2080TI, 64 GB memory) All editorial placeholders and typos (e.g., “Figure 2. This is a figure…” and batch size “62”) have been corrected.

Comment 9:

For future work, explore a lightweight visual–language pathway to condition decisions on textual metadata (FITS/WCS, observing conditions) and produce natural-language rationales. Such as: From Gaze to Insight: Bridging Human Visual Attention and Vision Language Model Explanation for Weakly-Supervised Medical Image Segmentation.

Response:

Thank you for your suggestion. We have added this point to the Future Work section, where we plan to explore a visual–language integration framework in future studies.

Round 2

Reviewer 2 Report (New Reviewer)

Comments and Suggestions for Authors

The paper presents a promising approach, Ensemble Deep Learning for Real–Bogus Classification with Sky Survey Images.

Authors are suggested to clarify the following comments:

  1.  Not enough recent reference works are included; please update them.

Author Response

Comment : Not enough recent reference works are included; please update them.

Response:

Thank you for your valuable suggestion. I have carefully revised the manuscript and increased the number of reference works from 34 to 57. The new references have been mainly added in the Materials and Methods section to strengthen the methodological foundation, and also in the Introduction section to include more recent studies related to astronomical research and transient detection. These updates ensure that the manuscript reflects the latest developments in the field and provides a more comprehensive context for the study.

This manuscript is a resubmission of an earlier submission. The following is a list of the peer review reports and author responses from that submission.

Round 1

Reviewer 1 Report

Comments and Suggestions for Authors

(1)Although transfer learning effectively leverages the feature extraction capabilities of pre-trained models, significant domain differences exist between the ImageNet dataset (which contains everyday objects such as cats, dogs, and cars) and astronomical images (which include point sources, nebulae, noise, etc.). These differences in low-level features may prevent the model from optimally capturing the unique and subtle characteristics of astronomical images, such as extremely faint transient signals or specific point spread function morphologies. Therefore, more intensive fine-tuning or specific architectural adjustments are required, and the final performance must be rigorously validated through experiments.

(2)The data augmentation techniques mentioned in the manuscript, e.g., rotation and flipping, primarily simulate variations in observation angle and orientation. However, real-world astronomical images exhibit far more complex variations, which may include atmospheric turbulence-induced changes in seeing conditions, varying night sky brightness, different signal-to-noise ratios, sensor defects, cosmic ray hits, and occlusions caused by other celestial objects. Current data augmentation strategies may be insufficient to fully capture these intricate observational variabilities. As a result, models may experience performance degradation when encountering unmodeled noise or artifacts.

(3) Deep learning models are often regarded as "black boxes" due to their lack of decision transparency. In fields such as astrophysics, where rigorous scientific discovery is paramount, it is essential to understand why a model classifies a particular source as a transient. The absence of interpretability mechanisms—such as feature visualizations or saliency maps—may prevent astronomers from fully trusting the model’s predictions and hinder the extraction of new physical insights from its errors.

(4)The success of the research critically depends on the quality of the annotated training data. If the training dataset contains labeling errors, class imbalance, or selection bias, the model may learn and even amplify these inaccuracies and biases. It is therefore essential to provide a detailed description of the dataset construction process for both training and testing, including data cleaning procedures, and to employ rigorous cross-validation methods to ensure the statistical robustness and generalization ability of the evaluation results.

(5)The authors should compare their method against a more diverse set of state-of-the-art baselines, particularly those specifically designed for astrophysics. This will help to better position the contribution of this work within the existing landscape.

(6)A more rigorous ablation study is crucial to understand the contribution of each key component of the proposed architecture (e.g., the novel attention module, the specific loss function). Reporting results after systematically removing or altering each component would provide valuable insights into what truly drives the performance.

Reviewer 2 Report

Comments and Suggestions for Authors

As I read this manuscript, I am constantly asking one simple question: what’s genuinely new here, and does the evidence match the claim? If the central idea is an 'augmentation-savvy' ensemble with a heavily weighted noise-specialist, then the paper should make that the spine of the narrative and prove it decisively: clean ablations that pit this design against a single strong model, stacking with a meta-learner, test-time augmentation voting, and a Bayesian/uncertainty-aware baseline, evaluated only on a true validation set, not the test set. I also need absolute confidence in the data hygiene. Splits must be created before any augmentation so near-duplicates can’t leak; label provenance and likely label-noise rates should be made explicit; and the held-out test must resemble the alert distribution you woll face in production, not a sanitised or augmentation-heavy mix.

Methodologically, choices that seem convenient need scientific justification. Converting small FITS stamps to large JPEGs risks losing photometric fidelity and inventing morphology, either defend it with controls or keep the data native. Be clear whether you use difference/science/reference triplets or a single frame, and how you normalise backgrounds.  Why that freeze/unfreeze policy, why did some runs collapse, and how do focal/class-balanced/logit-adjusted losses or domain/self-supervised pretraining change the image?

Report PR/ROC with operating points at very low FPR, show calibration and reliability, and break performance down by artifact type and SNR, plus robustness across seasons/cameras/surveys. 

The similarity (plagiarism) rate is also a bit high (24%).

With all I would like to reject the paper as in its current format.

Reviewer 3 Report

Comments and Suggestions for Authors

In this paper, the authors introduced a novel method for discovering astronomical transient events using ensemble deep learning techniques. They used Convolutional Neural Networks (CNNs) to classify real and fake astronomical images with inspiration from biological vision systems. Though this paper is interesting, the following points should be addressed.

-- In the abstract, it would be helpful to include specific metrics or results to substantiate the claims, such as the percentage improvement in classification.

-- In the introduction, the authors should consider strengthening the rationale for using bio-inspired techniques with evidence of recent developments in the field and the potential these techniques have for success.

-- In addition to the above point, the authors should clearly state what specific obstacles traditional methods face in identifying transient events.

-- The authors should conclude the introduction with a summary, in bullet form, of what the contributions of this paper are.

-- There should be a section plan at the end of the introduction part i.e, section 2 discusses... section 3 gives...

-- The literature review may benefit from discussing the most recent relevant literature, including the following recent studies: Advancing multi-label melt pool defect detection in laser powder bed fusion with self-supervised learning; Enhancing time-series prediction with temporal context modeling: a Bayesian and deep learning synergy; Unbiased normalized ensemble methodology for zero‐shot structural damage detection using manifold learning and reconstruction error from variational autoencoder; Novel integration of forensic-based investigation optimization algorithm and ensemble learning for estimating hydraulic conductivity of coarse-grained road materials.

-- The authors might consider including a brief discussion of the significance of each data augmentation technique they employ, and what they expect will happen to model performance.

-- It would be useful if the authors could somehow connect findings to potential real-world applications, like the role of improved detection methods in helping understand the behavior of cosmic phenomena.

-- If applicable, the authors might study the potential of incorporating Generative Adversarial Networks (GANs) to produce synthetic data and improve model training, especially for classes that are under-represented.

-- The authors might study the applicability of their methods for use on other astronomical datasets and how their methods performs and generalises in other contexts.

Reviewer 4 Report

Comments and Suggestions for Authors

After reviewing your article, I have come to the conclusion that it should not be accepted for the reasons stated below. The comments below outline the shortcomings and areas for improvement identified through a thorough examination of your work.

Lack of Model Training Graphs:

  • Your article does not present any graphs or visuals related to the model training processes. Furthermore, the analyses in the article were performed using traditional deep learning models. Instead, your proposed model could have been developed using more innovative hybrid approaches. The lack of graphs visualizing the model's learning process, training accuracy, loss function, and validation results makes it difficult to observe how effectively the model was trained and whether there are any overfitting or underfitting issues. Such visuals are critical, especially in evaluating deep learning models.

Lack of Confusion Matrix and Performance Metrics:

  • Your paper only uses high-level metrics such as overall accuracy rates and F1 scores to evaluate model performance. However, a more detailed evaluation is expected in classification problems. Specifically, using a confusion matrix provides more detailed information about the relationship between actual classes and predicted classes, as well as the model's misclassification tendencies. This information is crucial for understanding how the model actually performs.

Lack of Details on Data Augmentation and Training Processes:

  • There is a lack of clear discussion on how data augmentation is applied and the effect of each augmentation strategy on model performance. In particular, the effect of the specific data augmentation methods used in training each model (e.g., how each model responds to rotation, reflection, or noise addition techniques) needs to be reported more clearly. This will help us understand which data diversity each model performs better with and why.

Insufficient Explanation of the Ensemble Learning Strategy:

The ensemble learning strategy, particularly how the Weighted Voting and Soft Voting methods are applied, should be explained in more detail. Critical details are missing, such as which model is given more weight, how these weights are determined, and why the ensemble decision-making mechanism is structured this way. These explanations are important for understanding the effectiveness of the ensemble strategy and the reliability of the model.

Lack of Mathematical and Statistical Calculations:

  • Your article focuses on the metrics used for model performance, particularly accuracy and F1 score. However, the calculation processes for these metrics and, in particular, the techniques used to deal with imbalanced data need to be explained in more detail. In addition, statistical analyses of how class imbalance affects model performance would have been expected.

Reasons for and Effects of Conversion from FITS to JPG:

  • No analysis has been performed on whether converting the FITS file to JPEG causes data loss and how this conversion affects the model's accuracy. The FITS format typically stores high-quality data in astronomical images, so the results of this conversion should be discussed more clearly.

Insufficient Discussion and Generalizability of Results:

  • The conclusion section requires further discussion on how the proposed approach can be generalized to real-world astronomical datasets. Specifically, more information should be provided on the model's flexibility regarding different observation conditions, image quality, and light variations.

Conclusion and Rejection Decision:

The shortcomings mentioned above indicate that your paper has not been sufficiently comprehensive and in-depth, and that important points have been overlooked. For these reasons, I have rejected your paper. However, these criticisms highlight areas for improvement in future work, and a revision that addresses these shortcomings in the future may help you achieve greater success.